# Position: All Current Generative Fidelity and Diversity Metrics are Flawed

**Ossi Räisä** [1]   **Boris van Breugel** [2]   **Mihaela van der Schaar** [2]

## Abstract

Any method's development and practical application is limited by our ability to measure its reliability. The popularity of generative modeling emphasizes the importance of good synthetic data metrics. Unfortunately, previous works have found many failure cases in current metrics, for example lack of outlier robustness and unclear lower and upper bounds. We propose a list of desiderata for synthetic data metrics, and a suite of sanity checks: carefully chosen simple experiments that aim to detect specific and known generative modeling failure modes. Based on these desiderata and the results of our checks, we arrive at our position: all current generative fidelity and diversity metrics are flawed. This significantly hinders practical use of synthetic data. Our aim is to convince the research community to spend more effort in developing metrics, instead of models. Additionally, through analyzing how current metrics fail, we provide practitioners with guidelines on how these metrics should (not) be used.

## 1. Introduction

Recent years have seen great advances in generative modeling, from generating realistic facial images (Karras et al., 2019) to generating tabular datasets with large language models (Borisov et al., 2022). These advances are motivated by a promise that the synthetic data generated by these models could augment real data (Das et al., 2022), maintain privacy of sensitive data (Liew et al., 1985; Rubin, 1993), improve model evaluation (van Breugel et al., 2023), or improve fairness in downstream tasks (van Breugel et al., 2021), among some possible goals.

The quality of a generative model is typically evaluated using metrics that compare the real data with synthetic data produced by the model. Popular metrics include

---
[1] University of Helsinki [2] University of Cambridge. Correspondence to: Ossi Räisä <ossi.raisa@helsinki.fi>.

*Proceedings of the 42nd International Conference on Machine Learning*, Vancouver, Canada. PMLR 267, 2025. Copyright 2025 by the author(s).

FID (Heusel et al., 2017), information divergences like total variation distance and KL divergence, and the performance of downstream machine learning tasks.

However, while these metrics can distinguish good and bad generators, they do not provide information on why a particular generator received a poor evaluation, or what can still be improved in a good generator. Sajjadi et al. (2018) proposed a precision/recall metric that distinguishes two failure modes: a generator that generates unrealistic samples and a generator that does not cover all of the real distribution. These metrics were first improved by Kynkäänniemi et al. (2019) and later others, leading to several metrics, including density/coverage (Naeem et al., 2020) and $\alpha$-precision/$\beta$-recall (Alaa et al., 2022). The latter two pairs are the most popular of these types of metrics today.

These metrics come in pairs. The first metrics of the pairs measure how realistic synthetic samples are, so we call them *fidelity* metrics. The second metrics of each pair measure how much of the real distribution the synthetic distribution covers, so we call them *diversity* metrics.

While evaluating and improving generative models has received much attention, evaluating and improving the metrics themselves has received much less. Some recent works have found failure cases with the established fidelity and diversity metrics (Cheema & Urner, 2023; Khayatkhoei & Abdalmageed, 2023; Park & Kim, 2023), and proposed new metrics that fix the discovered problems. However, each of these works only looks at a small number of problems and focuses on fixing those. It remains an open question whether the fixed metrics suffer from the failure cases reported by the other works, or if the fixes cause new problems to emerge.

We aim to answer this question with a thorough evaluation of fidelity and diversity metrics, consisting of three main contributions:

1. We propose a list of six desiderata that any synthetic data metric should fulfill in Section 2.

2. We distill each failure case that has been reported in the literature into a simple *sanity check*, with precisely defined and automatically checked passing criteria, that are linked to the desiderata, in Section 4. Our imple-

mentation code is available.[1]

3. We evaluate whether each metric passes each check, and discuss the results in Section 5.

We also add some novel sanity checks that focus on tabular data, which is an important domain where these metrics are used (Kotelnikov et al., 2023; Zhang et al., 2023), but has been neglected by existing work on metrics.

The results of our evaluation in Tables 3 and 4 lead to our position. All of the fidelity and diversity metrics fail a large number of sanity checks, in many cases failing to measure even the basic property that they are supposed to measure, which we argue means that **all current generative fidelity and diversity metrics are flawed.**

We argue that our position has two main takeaways:

1. Practitioners using fidelity and diversity metrics to evaluate synthetic data must be wary of what the metrics they use can measure, and what they fail to measure.

2. New metrics are needed to fix the failure cases in the current ones. These new metrics must be evaluated in a wide range of scenarios to uncover potential failure cases, including a benchmark of sanity checks like the one we present.

The first takeaway is a consequence of the clear unreliability of all metrics in our sanity checks. A metric suffering from a failure case that is present in a given setting does not give useful information on synthetic data quality. This is complicated by the complexities of real data: while our sanity checks each look at one potential failure case at a time, a real dataset will likely have multiple potential failure cases at the same time. If a metric fails at one of these cases, it can also fail at the others, even if it does not fail on those in isolation.

The second takeaway follows from the first and the usefulness of the information fidelity and diversity metrics aim to measure. This information is useful for generative model evaluation, as evidenced by their wide-scale adoption in the community (Pearce et al., 2022; Anciukevičius et al., 2023; Kotelnikov et al., 2023; Qian et al., 2023; Zhang et al., 2023). Since the current metrics we have are not completely reliable, as the generative model research community, we must strive to develop reliable alternatives that can be used without worrying about a minefield of failure cases.

## 1.1. Related Work

Borji (2019; 2022); Xu et al. (2018) survey evaluation metrics with a focus on GANs in the image domain. Borji

---

[1] https://github.com/vanderschaarlab/position-fidelity-diversity-metrics-flawed

(2019); Xu et al. (2018) give lists of desiderata, which partially overlap with our list in Section 2, and ask which metrics fulfill each of the desiderata. However, their desiderata focus on GANs, and demand that a single metric distinguishes several generator qualities like fidelity, diversity, and overfitting at the same time. We consider it sufficient for a metric to focus on a single quality at a time. Theis et al. (2016) looks at several classical metrics and finds that they can result in conflicting and undesirable evaluations. Theis (2024) considers what properties a fidelity, or "realism" in their terminology, metric should have, and gives some theoretical insight on how a metric with those properties could be implemented.

## 2. Desiderata for Synthetic Data Evaluation Metrics

Before we can evaluate how good different metrics are, we must establish what we want out of a metric. We use the lists given by Borji (2019); Xu et al. (2018) as a starting point, but we do not require a single metric to evaluate multiple aspects of synthetic data quality. While we focus on fidelity and diversity metrics in our evaluation, we take a wider perspective in our desiderata list, and include goals that other types of metrics may aim to meet.

Our desiderata for synthetic data metrics are:

D1 (purpose) One of the following:

   (a) measure a quantity that is of direct practical interest,

   (b) give interpretable information on the difference between real and generated data,

   (c) be a proxy-metric that is correlated with an impractical metric of interest.

D2 (hyperparameters) Have a minimal number of hyperparameters. The effect of hyperparameters should be as clear as possible.

D3 (data) Require less data than what is available in the problem of interest.

D4 (bounds) Have clear lower and upper bounds.

D5 (invariance) Be invariant to transformations of both real and synthetic data that do not affect data quality in the domain of interest.

D6 (computation) Be computationally efficient.

These are not in any sort of priority order.

Examples of D1a (purpose) are the accuracy of predictive models or statistical inference on synthetic data. Fidelity and diversity metrics that are the focus of this paper are

examples of D1b. Many works evaluate metrics by looking at D1c, for example by comparing the metric values with human evaluations (Salimans et al., 2016; Jayasumana et al., 2024). We include the condition that the target metric is impractical in some way, or at least less practical than the proxy metric, since otherwise there is no need for the proxy metric. If the target metric can be practically computed, a practitioner could simple compute the target metric instead of a proxy metric that is merely correlated with it.

D2 (hyperparameters) aims to maximise the objectivity of synthetic data evaluation. A metric with many hyperparameters with unclear effects may have many seemingly equivalent choices for the hyperparameters, which nevertheless lead to different estimations of synthetic data quality. Hyperparameters with a standard value, or method of choosing the value, should not be counted here, since a default choice forbids the tweaking of the hyperparameter to obtain a desired result from a comparison.

For D3 (data), an ideal metric would be completely invariant to the size of the real data, but this is an unreasonable requirement for very small dataset sizes. Instead, we want the metric to be approximately invariant after some size, which we call the required size. The required size should of course be smaller than the amount of data that is available in a given problem of interest. For this paper, we set a threshold of 1000 datapoints, which is small enough to cover the majority of datasets[2], but large enough for many statistical asymptotics to kick in. Note that the size of the synthetic dataset is separate from the real data, though the two are often set to be identical. Since the only constraint for the size of the synthetic data is computational, we consider it a hyperparameter of the metric, and place it under desiderata D2 (hyperparameters).

D4 (bounds) allows gauging how good or bad the synthetic data is in absolute terms, not only in comparison to other synthetic datasets. This is important, since it is possible that in a hard setting, even the best synthetic data is of poor quality.

The transformations that do not affect data quality in D5 (invariance) are domain specific. For images, examples are small translation and rotations (Borji, 2019). For tabular data, we can identify three types of transformation that clearly do not affect data quality: scaling numerical variables, permuting the categories of nominal discrete variables, and permuting the order of variables. The latter two are clearly irrelevant transformations, since they are changes to orderings that are arbitrarily chosen in the first place. Scaling a numerical attribute is equivalent to changing units. Changes of units in both real and synthetic data

do not change the quality of synthetic data, so they should not change the values of evaluation metrics either.

A minimum threshold of computational efficiency for D6 (computation) is tractability: the metric should be possible to compute in reasonable time. Of course, computational efficiency is always useful for a metric, but for tractable metrics, the other desiderata should be prioritised over pure computational savings.

## 3. Evaluated Metrics

**Inclusion Criteria** We include metrics in our evaluation that:

1. measure fidelity or diversity,

2. produce a single number and

3. run in reasonable time.

Requirement (2) excludes the curve-valued metric of Sajjadi et al. (2018) and improvements to it (Simon et al., 2019; Djolonga et al., 2020; Siry et al., 2023; Sykes et al., 2024). We only include single number metrics since they are easier to interpret and display than curve-values ones, and are much more common in practice (Pearce et al., 2022; Anciukevičius et al., 2023; Kotelnikov et al., 2023; Qian et al., 2023; Zhang et al., 2023). We also looked at the pair of metrics from Kim et al. (2023) that are based on topological properties, but they took too much time to compute a single evaluation.[3] We list the metrics that we include in Table 1. We describe each metric in detail and list the implementations we used in Appendix A.

**Embeddings** All of the metrics we consider take a set of real data $X_r \sim P_r$ and a set of synthetic data $X_g \sim P_g$. They first embed $X_r$ and $X_g$ into a space with more suitable geometry than the original data space. We denote the embedding of a single real or synthetic datapoint as $\phi_r$ or $\phi_g$, respectively, and denote the embeddings of the whole real or synthetic datasets as $\Phi_r$ and $\Phi_g$, respectively.

For image datasets, the embedding is typically a pretrained neural network (Kynkäänniemi et al., 2019; Naeem et al., 2020), or a randomly initialised CNN (Naeem et al., 2020). For tabular data, Alaa et al. (2022) use a neural network embedding that is trained on the real data. With the other metrics, we will use a simpler embedding that is appropriate for tabular data: we one-hot encode categorical variables,

---

[2]As of 29th of January, 2025, 55% of datasets on `https://openml.org/` have $> 1000$ datapoints.

[3]Specifically, computing the metrics of Kim et al. (2023) for 1000 real and synthetic samples of 2-dimensional Gaussian distributions with the original implementation did not finish in 30 minutes on an M1 MacBook Air. Computing all of I-Prec, I-Rec, density and coverage in the same setting took less than 0.1s.

*Table 1.* Metrics in this work.

| Paper | Fidelity Metric | Diversity Metric |
|---|---|---|
| (Kynkäänniemi et al., 2019) | Improved Precision (I-Prec) | Improved Recall (I-Rec) |
| (Naeem et al., 2020) | Density | Coverage |
| (Alaa et al., 2022) | Integrated $\alpha$-precision (IAP) | Integrated $\beta$-recall (IBR) |
| (Cheema & Urner, 2023) | Precision Cover (C-Prec) | Recall Cover (C-Rec) |
| (Khayatkhoei & Abdalmageed, 2023) | Symmetric Precision (symPrec) | Symmetric Recall (symRec) |
| (Park & Kim, 2023) | Probabilistic Precision (P-Prec) | Probabilistic Recall (P-Rec) |

and normalise numerical variables to mean zero and unit standard deviation on the real data.[4]

This simple embedding is motivated by our desiderata D5 (invariance) and D2 (hyperparameters). Normalising numerical variables ensures that the resulting metric in invariant to scaling. One-hot encoding categorical variables ensures that the metric is invariant to permuting the order of categories, as long as the metric is invariant to permuting the components of the embedding vector $\phi$, which is the case for all of the metrics we look at. This also ensures that the metric is invariant to permuting the order of variables in the real data, so this embedding ensures invariance to all of the tabular data transformations that clearly do not affect data quality we discussed in Section 2. This embedding also does not have hyperparameters. In contrast, all architecture and training choices of a neural network embedding become hyperparameters of the resulting metric, unless there is a standard choice of them that is widely used.

## 4. Sanity Checks

To evaluate the failure cases each metric may or may not have, we use simple sanity checks, with artificial real and synthetic distributions. This allows us to focus on one potential problem in each check, and also allows evaluating extreme scenarios where we know that a metric meeting D4 (bounds) should have a value of 0 or 1. Most of our checks are from existing work, but we do not always use identical setups in these cases. A handful of checks are new to this work. Table 2 lists the sanity checks we look at and previous works using them. Appendix B provides detailed descriptions of each check and the setup we use. Next, we very briefly summarise each check.

### 4.1. Sanity Check Summaries

**Gaussian Checks** To evaluate how metrics distinguish simple differences between distributions, we have five sanity checks looking at two otherwise identical Gaussian distributions that differ in one way. In the first three checks, the difference is either their mean, their standard deviation, or their mean and the presence of an outlier. In the fourth and fifth check, the difference is their mean in one dimension. In the fourth check, there is one additional identical dimension, which is scaled, testing whether the metrics are invariant to the scaling. In the fifth check, there are many additional identical dimensions, testing whether metrics can find the difference in only one of the dimensions.

**Gaussian Mixture Checks** These checks evaluate how the metrics see mode collapse, invention and dropping. One check drops 9 out of 10 modes either one mode at a time, or by reducing the weight of all 9 dropped modes simultaneously. A second check evaluates both mode dropping and invention by increasing the number of modes in the synthetic distribution, first including modes in the real data, and later inventing new modes. A third check evaluates mode collapse using a real distribution with two modes, and a synthetic distribution with one wide component that covers both real modes.

**Varying Dataset Sizes on Uniform Hypercubes** These checks evaluate the effect of both the real and synthetic dataset size on two uniform distributions on partially overlapping hypercubes. One experiment varies both $|\Phi_r| = |\Phi_g|$, while one fixes $|\Phi_r|$ and varies $|\Phi_g|$. Cheema & Urner (2023) argue that the theoretically correct values for fidelity and diversity metrics in this case are the volume of the overlapping space, which we have set to 0.2. For D1b (purpose), we check that the metrics are close to this value, and for D2 (hyperparameters) or D3 (data), we check that the metrics simply converge after 1000 datapoints.

**Uniform Hypersphere Surface Check** This check looks at the problem that inspired the symPrecision and symRecall metrics (Khayatkhoei & Abdalmageed, 2023). Both real and synthetic distributions are uniform on the surface of a hypersphere, with different radii. The distributions are disjoint, so all metrics should return low values, but Khayatkhoei & Abdalmageed (2023) observed that in high dimensions, many metrics only behave correctly when the synthetic radius is either smaller or larger than the real radius, but not in the opposite case.

---

[4]We also apply this embedding before the neural network of Alaa et al. (2022) for $\alpha$-precision and $\beta$-recall.

*Table 2.* Sanity Checks considered in this work. The Tab. column marks checks intended for only tabular data, and the Fig. column contains the figure number of the full results in the Appendix. The results are summarised in Tables 3 and 4. The checks are grouped as in Section 4.1.

| SETTING | DESIDERATA | TAB. | FIG. | PAPERS |
|---|---|---|---|---|
| GAUSSIAN MEAN DIFFERENCE | D1B, D4 | | 6 | (NAEEM ET AL., 2020; ALAA ET AL., 2022; CHEEMA & URNER, 2023) |
| GAUSSIAN MEAN DIFFERENCE + OUTLIER | D1B, D4 | | 7, 8 | (NAEEM ET AL., 2020; ALAA ET AL., 2022; PARK & KIM, 2023) |
| GAUSSIAN STD. DEVIATION DIFFERENCE | D1B, D4 | | 9 | (PARK & KIM, 2023) |
| ONE DISJOINT DIM. + MANY IDENTICAL DIM. | D1B, D4 | | 20 | THIS WORK |
| SCALING ONE DIMENSION | D4, D5 | | 18 | THIS WORK |
| MODE COLLAPSE | D1B, D4 | | 17 | (ALAA ET AL., 2022) |
| MODE DROPPING + INVENTION | D1B, D4 | | 12 | (KYNKÄÄNNIEMI ET AL., 2019) |
| SEQUENTIAL / SIMULTANEOUS MODE DROPPING | D1B, D4 | | 10, 11 | (NAEEM ET AL., 2020) |
| HYPERCUBE, VARYING SAMPLE SIZE | D3, D1B | | 14 | (CHEEMA & URNER, 2023) |
| HYPERCUBE, VARYING SYN. SIZE | D2, D1B | | 15 | (CHEEMA & URNER, 2023) |
| HYPERSPHERE SURFACE | D1B, D4 | | 13 | (KHAYATKHOEI & ABDALMAGEED, 2023) |
| SPHERE VS. TORUS | D1B, D4 | | 16 | (CHEEMA & URNER, 2023) |
| DISCRETE NUM. VS. CONTINUOUS NUM. | D1B, D4 | ✓ | 21 | THIS WORK |
| GAUSSIAN MEAN DIFFERENCE + PARETO | D1B, D4 | ✓ | 19 | THIS WORK |

**Torus vs. Sphere Check**  This check from Cheema & Urner (2023) evaluates metrics on distributions with non-trivial geometry. One of the distributions is a uniform distribution on a sphere, and the other is on a torus that surrounds the sphere, but is disjoint from it. See Figure 4 in the Appendix for an illustration. We vary which of the two distributions is considered real and which is considered synthetic.

**Tabular Data Focused Checks**  We include two sanity checks that focus on issues found in tabular data. The first check repeats the Gaussian mean difference check, but with an additional Pareto-distributed random variable with an identical distribution in real and synthetic data. This tests how the metrics function in the presence of a heavy-tailed power law distribution, which are common in tabular data. The second check has data from a Gaussian distribution, and data from the same Gaussian that has been rounded to an integer, varying the scale of both distributions. This evaluates whether the metrics can distinguish a discrete numerical distribution from a continuous one, which is important since integer-valued variables are common in tabular data.

### 4.2. Success Criteria

We evaluate the success or failure of each metric on the sanity checks with precisely defined criteria that we check programmatically. We detail the criteria of each check in Appendix B.

In general, we connect the criteria to one of desiderata D1b (purpose), D2 (hyperparameters), D3 (data), D4 (bounds) or D5 (invariance), depending on the check. D1b is evaluated in almost every check, with criteria that require the metric to generally behave correctly, but does not require specific values in most cases. D4 is also evaluated in almost every check, with criteria checking that the metrics have values close to 0 or 1 in extreme cases. We include these more difficult criteria because making absolute evaluations of generative models, like "this model is good" requires a metric with clear bounds that the metric actually follows in practice. D3 is evaluated in the two checks looking at dataset sizes, and requires that the metrics behave consistently after 1000 datapoints. D5 is evaluated in the scale invariance check, and checks whether the metrics are scale invariant. We do not include specific checks for D6 (computation), since all of the metrics we include are computationally tractable, so they pass the only sanity check that would make sense for D6.

**High and Low Diversity Metrics**  In some cases for diversity metrics, it can be argued that both high and low values are acceptable results, depending on how "covering" the real distribution is interpreted. These are cases where the synthetic distribution covers the real distribution, but is so wide that the probability (density) of getting a value from the synthetic distribution with high probability (density) under the real distribution is very low. An example of a sanity check like this is the two Gaussians with different standard deviations, with the synthetic standard deviation being much larger. On one hand, the synthetic distribution completely covers the real distribution, which suggests that diversity metrics should be high. On the other hand, the probability of sampling a synthetic datapoint in the high-density region of the real distribution is low, which suggests that diversity metrics should have low values. We designate diversity metrics that adhere to the former argument as "high", and designate metrics following the latter argument as "low". We require each metric to be consistent with "low" or "high" across different variations of a sanity check. We make these explicit in the results, so if only one of these viewpoints is valid in a given application, it is clear which metrics are suited to that application.

*Table 3.* Passes (T) and fails (F) of each fidelity metric on each sanity check. The Tab. column marks checks intended for only tabular data.

| Desiderata | Sanity Check | Tab. | I-Prec | Density | IAP | C-Prec | symPrec | P-Prec |
|---|---|---|---|---|---|---|---|---|
| | Discrete Num. vs. Continuous Num. | ✓ | F | F | F | F | F | F |
| | Gaussian Mean Difference | | T | T | T | T | T | T |
| | Gaussian Mean Difference + Outlier | | F | T | T | T | F | T |
| | Gaussian Mean Difference + Pareto | ✓ | T | T | T | T | T | T |
| | Gaussian Std. Deviation Difference | | T | T | F | F | F | T |
| | Hypercube, Varying Sample Size | | F | F | F | F | F | F |
| | Hypercube, Varying Syn. Size | | F | F | F | F | F | F |
| D1b (purpose) | Hypersphere Surface | | F | F | T | F | T | F |
| | Mode Collapse | | T | T | T | T | T | T |
| | Mode Dropping + Invention | | T | T | F | F | F | T |
| | One Disjoint Dim. + Many Identical Dim. | | F | F | F | F | F | F |
| | Sequential Mode Dropping | | T | T | F | F | F | T |
| | Simultaneous Mode Dropping | | T | F | F | F | F | T |
| | Sphere vs. Torus | | T | T | T | F | T | T |
| D2 (hyperparameters) | Hypercube, Varying Syn. Size | | T | T | T | F | F | T |
| D3 (data) | Hypercube, Varying Sample Size | | F | F | F | F | F | F |
| | Discrete Num. vs. Continuous Num. | ✓ | F | F | F | F | F | F |
| | Gaussian Mean Difference | | F | T | F | T | F | F |
| | Gaussian Mean Difference + Outlier | | F | F | F | T | F | F |
| | Gaussian Mean Difference + Pareto | ✓ | F | T | T | T | T | F |
| | Gaussian Std. Deviation Difference | | F | F | F | F | F | F |
| | Hypersphere Surface | | F | F | F | F | T | F |
| D4 (bounds) | Mode Collapse | | F | T | F | T | F | F |
| | Mode Dropping + Invention | | T | T | F | F | F | F |
| | One Disjoint Dim. + Many Identical Dim. | | F | F | T | F | F | F |
| | Scaling One Dimension | | F | T | T | T | T | T |
| | Sequential Mode Dropping | | F | T | F | F | F | F |
| | Simultaneous Mode Dropping | | F | T | F | F | F | F |
| | Sphere vs. Torus | | T | T | F | F | T | T |
| D5 (invariance) | Scaling One Dimension | | F | T | T | T | T | T |

## 4.3. Results

The passes and fails of each metric on each check are shown in Table 3 for fidelity metrics and Table 4 for diversity metrics. Plots showing the full results are included in Appendix C. We discuss the implications of these results in the next section.

## 5. Discussion

In this section, we discuss the main takeaways of our results. We start by looking at practical considerations in Section 5.1, and consider future research directions in Section 5.4.

### 5.1. Implications for Practitioners

Next, we go through the most important practical takeaways from our results. We present a checklist summarising these, along with more specific takeaways, in Section 5.2.

**No Metric is Suitable for Absolute Evaluations**    By absolute evaluation of a generative model, we mean answering the question "Is this model good / bad?". Research papers of-ten focus on relative evaluation, that is the question "Which model is the best?", however, practical deployments require a model that is good in absolute terms, not just relative to other models.

Absolute evaluation requires a metric with clearly defined lower and upper bounds, which can be chosen to be 0 and 1 without loss of generality. While most of the metrics we examine have these bounds in theory, our results in checks of D4 (bounds) show that they often do not have them in practice. For example, in Figures 6 to 9 with $d = 64$, an integrated $\alpha$-precision of around 0.5 is the best one can get, but with $d < 64$, 0.5 indicates bad synthetic data.

Another example is density, which behaves strangely in some cases. In most settings, the best density is close to 1, but in Figures 9 and 13 in high dimensions, density goes up to around 200. However, in these cases with identical distributions, density is around 1, so the high values do not correspond to better synthetic data.

**Investigate Effect of Real Dataset Size on Evaluation** The results of our D3 (data) check in Figure 14 show that all metrics are affected by the dataset size, many even at

*Table 4.* Passes (T, H or L) and fails (F) of each diversity metric on each sanity check. H and L refer to high and low metric types described in Section 4.2. The Tab. column marks checks intended for only tabular data.

| Desiderata | Sanity Check | Tab. | I-Rec | Coverage | IBR | C-Rec | symRec | P-Rec |
|---|---|---|---|---|---|---|---|---|
| | Discrete Num. vs. Continuous Num. | ✓ | F | F | F | F | F | F |
| | Gaussian Mean Difference | | T | T | T | T | T | T |
| | Gaussian Mean Difference + Outlier | | F | T | T | T | T | T |
| | Gaussian Mean Difference + Pareto | ✓ | T | T | T | T | T | T |
| | Gaussian Std. Deviation Difference | | H | F | L | F | F | H |
| | Hypercube, Varying Sample Size | | F | F | F | F | F | F |
| | Hypercube, Varying Syn. Size | | F | F | F | F | F | F |
| D1b (purpose) | Hypersphere Surface | | F | F | F | F | T | F |
| | Mode Collapse | | F | F | F | F | F | F |
| | Mode Dropping + Invention | | T | F | F | F | F | T |
| | One Disjoint Dim. + Many Identical Dim. | | F | F | F | F | F | F |
| | Sequential Mode Dropping | | T | T | F | F | T | T |
| | Simultaneous Mode Dropping | | F | T | F | T | F | T |
| | Sphere vs. Torus | | F | F | T | F | F | F |
| D2 (hyperparameters) | Hypercube, Varying Syn. Size | | F | F | F | F | F | F |
| D3 (data) | Hypercube, Varying Sample Size | | F | F | F | F | F | F |
| | Discrete Num. vs. Continuous Num. | ✓ | F | F | F | F | F | F |
| | Gaussian Mean Difference | | F | T | F | T | F | F |
| | Gaussian Mean Difference + Outlier | | F | T | F | T | F | F |
| | Gaussian Mean Difference + Pareto | ✓ | T | T | F | T | T | F |
| | Gaussian Std. Deviation Difference | | F | F | F | F | F | F |
| | Hypersphere Surface | | F | F | F | F | T | F |
| D4 (bounds) | Mode Collapse | | F | T | F | T | F | F |
| | Mode Dropping + Invention | | T | F | F | F | F | F |
| | One Disjoint Dim. + Many Identical Dim. | | F | F | T | F | F | F |
| | Scaling One Dimension | | T | T | T | T | T | T |
| | Sequential Mode Dropping | | F | T | F | T | F | F |
| | Simultaneous Mode Dropping | | F | T | F | T | F | F |
| | Sphere vs. Torus | | T | F | T | T | F | T |
| D5 (invariance) | Scaling One Dimension | | F | T | T | T | T | T |

sizes around 10000. In many benchmarks, there are choices that affect the dataset size, such as the handling of missing values, or the choice of a subset from a larger dataset. Since these choices also affect the metrics through the dataset size, one should investigate how large this effect is in any given evaluation, and whether it could affect the conclusions of the evaluation.

**Using Metrics with Real Data Requires Care** A real dataset is much more complex than any of our checks, and will likely have many of the potential failure cases that the checks look at simultaneously. If one uses a metric that fails in the presence of any one of these cases, the metric could fail as a whole and not provide any meaningful evaluation. As a result, it if one wants to use a metric in a particular setting, one should be able to justify that none of the cases the metric fails on are present in the setting, or otherwise justify that the failures of the metric are not a concern in the setting.

> **Takeaway for Practitioners**
>
> Practitioners using fidelity and diversity metrics must be wary that:
>
> - No metric is suitable for absolute evaluations.
>
> - The effect of dataset size on metrics should be investigated in each evaluation.
>
> - Using metrics with real data requires care, see checklist in Section 5.2.

### 5.2. Practical Advice Checklist

This checklist summarises the main takeaways from our results for practical applications of fidelity and diversity metrics. The list is intended for practitioners designing an experiment evaluating generative models. For each experiment, the designer should answer the questions, and, if advice is given for their answer, take that into account in their design. If there are multiple preferred metrics after answering all questions, we encourage computing all of them

to see whether they all lead to the same conclusion, or to different conclusions.

1. Is your evaluation absolute (is the model good) or relative (which model is best)?

    • Absolute: all current metrics are bad

2. Does the size of your evaluation dataset depend on choices you have made during data preparation, such as how missing data is handled?

    • Yes: Investigate how the metrics you are using behave with different dataset sizes, and take this into account when interpreting your results.

3. Is detecting mode collapse / dropping / invention important?

    • Yes: prefer I-Prec, P-Prec and P-Rec

4. Is robustness to outliers important?

    • Yes: avoid I-Prec, symPrec and I-Rec

5. Is it important to find differences in one dimension among many?

    • Yes: all current metrics are bad

6. Is it important to find a difference between continuous and discrete numerical values?

    • Yes: all current metrics are bad

7. Is the distinguishing between distributions on close surfaces important (uniform hypersphere surface check)?

    • Yes: prefer symPrec, IAP and symRec

8. Is it important to distinguish distributions on complex shapes, with possibly surrounding the other (sphere vs. torus check)?

    • Yes: avoid C-Prec among fidelity metrics, prefer IBR among diversity metrics

**Reasons for Checklist Advice**   The first two questions on the checklist are discussed in Section 5.1. The advice in the checklist for these questions summarise the recommendations from the corresponding discussion. For the rest of the questions, we look at relevant checks in Tables 3 and 4, and recommend metrics that pass all or most relevant checks, and recommend avoiding metrics that fail many relevant checks. In general, we avoid giving any hard recommendations of a particular metric due to the fact that there is no clearly best metric in any case.

## 5.3. Hardest Sanity Checks

In this section, we examine the hardest sanity checks that most or all metrics failed, and discuss possible reasons for the failures.

**One Disjoint Dimension**   This check is about finding a large difference in one dimension among many identical dimensions. All metrics failed, and only $\alpha$-precision and $\beta$-recall are even close to passing in Figure 20. We conjecture that their partial success is due to the neural network embedding they use, which proved capable of learning an embedding where the difference in one dimension is apparent even with a large number of identical dimensions. The other metrics are based on Euclidean distance, where the large difference in one dimension gets drowned out by the other dimensions.

**Uniform Hypercube**   These checks test how the metrics behave as the sizes of the real and synthetic datasets changes. The real and synthetic distributions are uniform distributions on two overlapping $d$-dimensional hypercubes. All metrics fail to even converge, as required for D3 (data), when the sizes of both datasets change. When only the synthetic dataset size changes, many metrics are almost invariant, or at least converge reasonably, but none are close to the theoretical fidelity and diversity values from Cheema & Urner (2023). We conjecture that this is due to a non-intuitive feature of high-dimensional geometry. If the length of the overlapping sides between the two hypercubes were fixed, the overlap volume would decrease exponentially with dimension, so to keep the volume fixed, the overlapping length must quickly approach 1.[5] The distance between points that are a fixed length apart in all dimensions only increases polynomially with increased dimension, so the distances between the points in the real and synthetic hypercubes decrease with increased dimension in this setting.

**Hypersphere Surface**   This check evaluates a problem discovered by Khayatkhoei & Abdalmageed (2023) that motivated their symPrecand symRec metrics. However, this still remains an open problem, since those are the only metrics that pass this check, but they fail many basic D1b (purpose) checks. Future work should consider if the insights from Khayatkhoei & Abdalmageed (2023) on this problem can be applied to a fix with other metrics.

**Discrete vs. Continuous Numerical Variables**   This check evaluates whether the metrics can distinguish a continuous numerical distribution from a similar rounded distribution with discrete values. This was especially hard for current metrics, since they all failed. However, most of the failures happen only at a very fine discretisation, as seen in

---

[5]Both hypercubes have side length 1.

Figure 21.

### 5.4. Future Directions for Metrics

Since all metrics failed so many sanity checks that using any of them requires great care, it is clear that new fidelity and diversity metrics are needed that avoid as many of these failure cases as possible.

**Going Beyond Euclidean Geometry**   Many of the hard checks we have highlighted in Section 5.3 have a common theme: the check requires distinguishing things that are not easily distinguished in Euclidean geometry. As all of the metrics we evaluated are based on Euclidean distances in some way,[6] it is not surprising that they fail these checks. We conjecture that future work must go beyond Euclidean distances in some way, such as by using topological features (Barannikov et al., 2021; Southern et al., 2023; Kim et al., 2023), to solve these issues.

> **Takeaway for Researchers**
>
> More research on fidelity and diversity metrics is needed. This research should be evaluated on a wide range of sanity checks that aim to reveal as many failure cases as possible.

## 6. Alternative Views

**Sanity checks are unrealistic since they do not use real data**   All of our sanity checks use very simple distributions, so they are toy settings that lack the complexities of real data. As a result, one may argue that these checks are not representative of real settings, and are thus not useful to evaluate how metrics perform in practice.

We use simple artificial distributions in our sanity checks to isolate a single property that is tested in each check. This also allows us to set up extreme scenarios where we know what value a metric should have to meet D4 (bounds), which would be much harder with real data.

In addition, our position is that all current metrics are flawed. We argue that even simple artificial sanity checks are enough to show this, since a metric that does not work even in many simple settings in unlikely to work in more complex settings.

**Sanity check x/y/z is not relevant or is flawed**   Not all of our checks are relevant in all settings, and it is possible that there are flaws in their design or implementation, despite our efforts to make sure all the tests are correctly implemented and their criteria make sense. However, all metrics fail

---

[6]$\alpha$-precision and $\beta$-recall have learned non-Euclidean embedding, but still use Euclidean distances after applying the embedding.

much more than just one check. Arguing that some metric would pass all, or almost all, checks if irrelevant and flawed ones were removed would require there to be many of these removed checks, which we find very unlikely.

**A metric does not need to be perfect to be useful**   It is true that an imperfect metric can be very useful, and perfect metrics may not even exist at all. As a result, one may worry that our sanity checks raise a too high of a bar: reviewers may be inclined to reject papers proposing new metrics if those metrics fail any checks, even though the new papers might have genuinely useful insights. Setting the bar too high in this way could stall the development of metrics.

It is not our intention to claim that a metric must pass all of our checks to be useful. However, users of metrics must still know if if their metrics are imperfect, and how the flaws could affect their use case. In addition, as we argue in Section 5.1, a metric that fails on any potential failure case that is present in a setting could cause it to not provide any useful information, so the many failures of all current metrics means that their use needs to be considered very carefully.

## 7. Conclusion

In this work, we evaluated generative fidelity and diversity metrics using simple sanity checks that we collected from the literature, or proposed ourselves. Unlike the previous literature, we defined precise passing criteria for each check, allowing checking the criteria programmatically.

We found that all of the metrics fail many of the sanity checks, which lead to our position: all current fidelity and diversity metrics are flawed in one way or the other. We presented two takeaways from this position. First, practitioners using these metrics must be wary of their flaws, and second, more research into metrics is needed to correct the flaws of the current ones. We presented several alternative views to our position, and gave our arguments against each alternative.

We hope to encourage better informed use of fidelity and diversity metrics in future works, and to highlight the need for new metrics with less potential issues. We think that these will lead to more accurate evaluations of generative models, and help continue the great advances in generative modeling of recent years into the future.

### Acknowledgements

OR was supported by the Researchers Abroad (Tutkijat maailmalle)-program (project 20240109).

## Impact Statement

The goal of this paper is to encourage informed use of fidelity and diversity metrics when evaluating generative models, and the development of better metrics. We think the societal consequences of this work are likely to be positive, since naive use of poor metrics may lead to incorrect conclusions about the models the metrics are measuring.

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

# A. Metrics in This Paper

**Notation** Recall that all metrics we consider are computed on embeddings of datapoints. We denote the embedded value of a datapoint by $\phi$, and denote an embedded dataset by $\Phi$. We differentiate real and generated data with subscripts: $\Phi_r$ is real data, $\Phi_g$ is generated data. We use $|\Phi|$ to denote the size of the dataset $\Phi$.

**Source Code Repositories** We used to following source code repositories for the metrics in this work:

- I-Prec, I-Rec, Density, Coverage: `https://github.com/clovaai/generative-evaluation-prdc`, MIT License

- IAP, IBR: `https://github.com/ahmedmalaa/evaluating-generative-models`, MIT license or BSD 3-clause license (repository has MIT, files say BSD 3-clause)

- C-Prec, C-Rec: `https://github.com/FasilCheema/GenerativeMetrics`, GPL-3.0 license

- symPrec, symRec: `https://github.com/mahyarkoy/emergent_asymmetry_pr`, MIT license

- P-Prec, P-Rec: `https://github.com/kdst-team/Probablistic_precision_recall`, MIT license

**Improved Precision / Recall** Kynkäänniemi et al. (2019) motivate the improved precision / recall metrics as improvements to the curve-valued metric of Sajjadi et al. (2018). In particular, improved precision and recall are single numbers, making comparisons with them easier, and they come with a practical algorithm to compute the metrics.

The basis for both improved precision and recall is approximating the support of the real or synthetic data distribution with a set of hyperspheres around each point, with radius set to the $k$th nearest neighbour of each point.

$$S(\Phi) = \bigcup_{\phi \in \Phi} B(\phi, \text{NND}_k(\phi, \Phi)) \tag{1}$$

where $B(\phi, r)$ is a hypersphere of radius $r$ centered at $\phi$, and $\text{NND}_k(\phi, \Phi)$ is the distance to the $k$th nearest neighbour[7] of $\phi$ in $\Phi$. Improved precision then simply counts the fraction of synthetic points that are in the support of the real data, while improved recall counts real points that are in the support of the synthetic data.

$$\text{I-Prec}(\Phi_r, \Phi_g) = \frac{1}{|\Phi_g|} \sum_{\phi_g \in \Phi_g} \mathbb{I}_{\phi_g \in S(\Phi_r)}, \tag{2}$$

$$\text{I-Rec}(\Phi_r, \Phi_g) = \frac{1}{|\Phi_r|} \sum_{\phi_r \in \Phi_r} \mathbb{I}_{\phi_r \in S(\Phi_g)}. \tag{3}$$

The hyperparameters of the metric are $k$, and the size of the synthetic dataset $|\Phi_g|$. Kynkäänniemi et al. (2019) choose $k = 3$ and use $|\Phi_r| = |\Phi_g| = 50000$ for image data. We follow their choice of $k$.

**Density / Coverage** Naeem et al. (2020) point out that improved precision and recall are vulnerable to outliers, since the hypersphere around an outlier is likely to be large, and the metrics simply count how many points are in at least one hypersphere. To fix these, Naeem et al. (2020) propose density and coverage.

Density counts, for each synthetic datapoint, how many hyperspheres that the synthetic datapoint belongs to, and normalises the sum over all synthetic datapoints.

$$\text{Density} = \frac{1}{k|\Phi_g|} \sum_{\phi_g \in \Phi_g} \sum_{\phi_r \in \Phi_r} \mathbb{I}_{\phi_g \in B(\phi_r, \text{NND}_k(\Phi_r))}. \tag{4}$$

Coverage counts the fraction of real datapoints that have at least one synthetic point in their hypersphere.

$$\text{Coverage} = \frac{1}{|\Phi_r|} \sum_{\phi \in \Phi_r} \mathbb{I}_{\exists \phi_g \in \Phi_g \,:\, \phi_g \in B(\phi_r, \text{NND}_k(\phi_r, \Phi_r))}. \tag{5}$$

---

[7]Excluding $\phi$ itself.

The hyperparameters for density and coverage are $k$ and $|\Phi_g|$. Naeem et al. (2020) choose them according to an analytical expression for $\mathbb{E}[\text{coverage}]$ for identical real and synthetic distributions. They first set $|\Phi_r| = |\Phi_g| = 10000$, and then set $k = 5$ to obtain $\mathbb{E}[\text{coverage}] > 0.95$. Since we do not always use $|\Phi_r| = |\Phi_g| = 10000$, we solve $k$ from the expression of $\mathbb{E}[\text{coverage}]$ given by (Naeem et al., 2020, Lemma 2), which supports $|\Phi_r| \neq |\Phi_g|$. We set a maximum $k$ of 20.

**$\alpha$-precision / $\beta$-recall**  Instead of placing hyperspheres around each datapoint like the previous metrics, $\alpha$-precision and $\beta$-recall estimate the supports with a single hypersphere that contains an $\alpha$ (or $\beta$) fraction of the datapoints (Alaa et al., 2022). Using a single hypersphere to approximate the support clearly requires the embedding space to have suitable geometry, which is why Alaa et al. (2022) use a DeepSVDD neural network (Ruff et al., 2018), which explicitly attempts to embed the datapoints inside a hypersphere, as their embedding. They train this network on the real data in tabular settings.

Let $S_\alpha(\Phi_r)$ be the minimum-volume hypersphere that contains an $\alpha$ fraction of the real embeddings. Then

$$P_\alpha = \frac{1}{|\Phi_g|} \sum_{\phi_g \in \Phi_g} \mathbb{I}_{\phi_g \in S_\alpha(\Phi_r)}. \tag{6}$$

Let $S_\beta(\Phi_g)$ be the minimum-volume hypersphere that contains a $\beta$-fraction of the synthetic embeddings. Let

$$\phi_{g,\beta}^* = \text{NN}_1(\phi_r, \Phi_g \cap S_\beta(\Phi_g)) \tag{7}$$

where $\text{NN}_1(\phi, \Phi)$ is the nearest neighbour[8] of $\phi$ in $\Phi$. Then

$$R_\beta = \frac{1}{|\Phi_r|} \sum_{\phi_r \in \Phi_r} \mathbb{I}_{\phi_{g,\beta}^* \in B(\phi_r, \text{NND}_k(\phi_r, \Phi_r))}. \tag{8}$$

$P_\alpha$ and $R_\beta$ have a separate value for each $\alpha, \beta \in [0, 1]$. To aggregate these values, Alaa et al. (2022) define integrated $\alpha$-precision and $\beta$-recall as

$$\text{IAP} = 1 - 2 \int_0^1 |P_\alpha - \alpha| \mathrm{d}\alpha, \tag{9}$$

$$\text{IBR} = 1 - 2 \int_0^1 |R_\beta - \beta| \mathrm{d}\beta. \tag{10}$$

These integrated metrics are what we evaluate in our experiments.

$\alpha$-precision and $\beta$-recall have the architecture choices and training details of the embedding network as hyperparameters. $\beta$-recall additionally has $k$. We use the code of Alaa et al. (2022) to compute both metrics, so we use their choices for the embedding network and $k$. We use the same normalisation we use with other metrics before applying the neural network embedding.

**Precision / Recall Cover**  Precision cover and recall cover (Cheema & Urner, 2023) are based on the intuition that metrics should not care whether a part of the support is covered if that part has a very small probability. For example, parts of the real data distribution that have very low probability need not be covered by the synthetic data generator.

$$\text{C-Prec} = \frac{1}{|\Phi_g|} \sum_{\phi_g \in \Phi_g} \mathbb{I}_{|\{\phi_r \in \Phi_r | \phi_r \in B(\phi_g, \text{NND}_{k'}(\phi_g, \Phi_g))\}| \leq k}, \tag{11}$$

$$\text{C-Rec} = \frac{1}{|\Phi_r|} \sum_{\phi_r \in \Phi_r} \mathbb{I}_{|\{\phi_g \in \Phi_g | \phi_g \in B(\phi_r, \text{NND}_{k'}(\phi_r, \Phi_r))\}| \leq k}. \tag{12}$$

Precision cover and recall cover have the hyperparemeters $k$ and $k'$. Cheema & Urner (2023) set $k' = \lceil \ln |\Phi_r| + 6 \rceil$ and $k = \lceil k'/3 \rceil$, which we follow in our experiments.

---

[8]Including $\phi$ itself, so if $\phi \in \Phi$, $\text{NN}_1(\phi, \Phi) = \phi$.

**Symmetric Precision / Recall**    Khayatkhoei & Abdalmageed (2023) point out that previous metrics behave asymmetrically when the real distribution is on the surface of a hypersphere, and the synthetic distribution is on the surface of another hypersphere with the same center, but either a larger or smaller radius. The asymmetric behavior is a problem, since the surfaces of the hyperspheres are disjoint, so both fidelity and diversity measures should be close to zero when the difference in radii is large enough.

To fix the asymmetry, Khayatkhoei & Abdalmageed (2023) first define complement versions of improved precision and recall

$$\text{cPrecision} = \frac{1}{\Phi_g} \sum_{\phi_g \in \Phi_g} \mathbb{I}_{|B(\phi_g, \text{NND}_k(\phi_g, \Phi_g)) \cap \Phi_r| \geq 1}, \tag{13}$$

$$\text{cRecall} = \frac{1}{\Phi_r} \sum_{\phi_r \in \Phi_r} \mathbb{I}_{|B(\phi_r, \text{NND}_k(\phi_r, \Phi_r)) \cap \Phi_g| \geq 1}. \tag{14}$$

$$\tag{15}$$

cPrecision counts the fraction of synthetic data points whose $k$-nearest neighbourhoods contain at least one real datapoint, and cRecall reverses the roles of real and synthetic data. Note that cRecall is the same quantity as coverage.

cPrecision and cRecall have the problematic asymmetric behaviour, but in opposite direction from improved precision and recall, so Khayatkhoei & Abdalmageed (2023) define symmetric metrics that fix the asymmetry.

$$\text{symPrec} = \min(\text{cPrecision}, \text{I-precision}), \tag{16}$$

$$\text{symRec} = \min(\text{cRecall}, \text{I-Recall}). \tag{17}$$

The hyperparameters for symPrecision and symRecall are $k$, and $|\Phi_g|$. Khayatkhoei & Abdalmageed (2023) use $|\Phi_g| = |\Phi_r| = 10000$, and set $k = 5$. We follow their choice of $k$.

**Probabilistic Precision / Recall**    Park & Kim (2023) aim to solve the outlier robustness problem of improved precision and recall by approximating the supports of the real and synthetic data probabilistically. Instead of computing a binary value of whether a datapoint is in the approximate support or not, they compute compute an estimate for the probability that the point is in the support, and average these probability estimates to obtain the aggregate metrics called probabilistic precision and recall.

Park & Kim (2023) call the function estimating the probability of $\phi$ to be in the support of $\Phi$ the probabilistic scoring rule (PSR), and define it as

$$\text{PSR}_\Phi(\phi') = 1 - \prod_{\phi \in \Phi} (1 - f(\phi, \phi', R_\Phi)). \tag{18}$$

The function $f(\phi, \phi', R_\Phi)$ is a simple estimate of the probability that $\phi'$ is in the support around $\phi$:

$$f(\phi, \phi', R) = \begin{cases} 1 - \frac{||\phi - \phi'||_2}{R} & \text{if } ||\phi - \phi'||_2 \leq R \\ 0 & \text{otherwise} \end{cases} \tag{19}$$

where

$$R_\Phi = \frac{a}{|\Phi|} \sum_{\phi \in \Phi} \text{NND}_k(\phi, \Phi) \tag{20}$$

and $a > 0$ is a hyperparameter.

Probabilistic precision and recall are then averages of the estimated probabilities:

$$\text{P-Prec} = \frac{1}{|\Phi_g|} \sum_{\phi \in \Phi_g} \text{PSR}_{\Phi_r}(\phi_g) \tag{21}$$

$$\text{P-Rec} = \frac{1}{|\Phi_r|} \sum_{\phi \in \Phi_r} \text{PSR}_{\Phi_g}(\phi_r). \tag{22}$$

The hyperparameters of P-precision and P-recall are $a$, $k$, which control the choice of $R$, and $|\Phi_g|$. Park & Kim (2023) set $a = 1.2$, $k = 4$ and $|\Phi_g| = |\Phi_r| = 10000$. We follow their choice of $a$ and $k$.

# B. Sanity Check Details

## B.1. General Success Criteria

We check the whether each metric passed or failed each check programmatically, checking whether the values of the metric meet given criteria. We detail these criteria when introducing each sanity check later in this section. Each criteria looks at specific points on the curves we plot in Section C, see for example Figure 6. The curves represent the mean metric values over 10 repeats. Some criteria only look at the value at a few significant points, often the left and right extremes and the middle. With these criteria, we check that the metric value at the given point is within 0.05 of a given value, unless otherwise specified. More complex criteria look at the overall shape of the curve by comparing values at several points. When there are multiple plots for different variations of the setting, such as multiple values of $d$ in Figure 6, a metric must pass the criteria for all variations. Next, we give the our detailed definitions of the overall shape criteria.

**Bell shape**     Informally, a bell-shaped curve has low values on the left, increases until a peak at a given midpoint, and then decreases again towards the right. Our specific criteria for a bell-shaped curve are:

- The difference between the values at both left and right extremes and at the midpoint is at least 0.2.

- The difference between the midpoint value and the maximum value is at most 0.1.

- The difference between the left extreme value or the right extreme value and the minimum value is at most 0.1.

**Low-to-high shape**     Informally, a low-to-high shape is any curve that has the lowest values at the left extreme and highest values at the right extreme, with a clear difference between the left and right extremes. Our specific criteria are:

- The difference between the right extreme value and the left extreme value is at least 0.2.

- The difference between the left extreme value and the minimum value is at most 0.1.

- The difference between the maximum value and the right extreme value is at most 0.1.

**High-to-low shape**     Informally, high-to-low shape is similar to the low-to-high shape, but the values go from high on the left to low on the right. Our specific criteria are:

- The difference between the left extreme value and the right extreme value is at least 0.2.

- The difference between the right extreme value and the minimum value is at most 0.1.

- The difference between the maximum value and the left extreme value is at most 0.1.

**High-to-low shape with middle drop**     Informally, this shape requires a curve to clearly decrease between the left and right extremes like the high-to-low shape, but also clearly decrease at a given point in the middle. Our specific criteria are:

- All of the criteria for a high-to-low shape.

- The difference between the left extreme value and a value at a given quantile of the x-axis is at least 0.1.

**Horizontal line shape**     This shape represents an approximately constant curve. Our specific criterion is that the difference between the maximum value and the minimum value is at most 0.05.

**Converging line shape**     This shape represents a curve that converges towards some value, and is then approximately constant. The convergence must happen before a given quantile of the x-axis. The specific criterion is that the difference between the maximum and minimum values that are to the right of the given quantile x-axis quantile is at most 0.05.

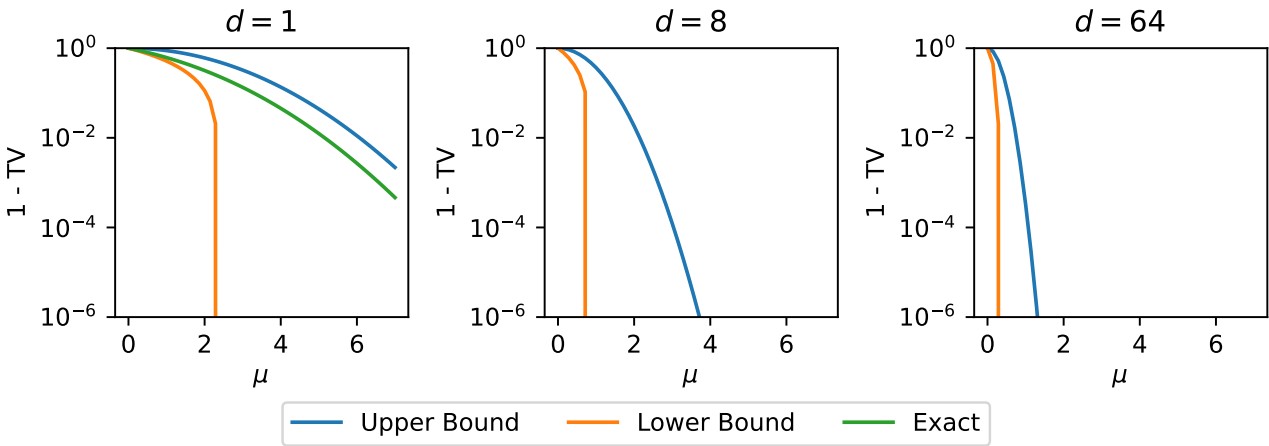

*Figure 1.* Total variation distance between a standard Gaussian and a Gaussian with mean $\mu 1_d$. The lower and upper bounds are based on Hellinger distance.

## B.2. Gaussian Mean Difference (+ outlier)

In this check, the real distribution is a $d$-dimensional Gaussian standard Gaussian distribution, and the synthetic distribution is a similar Gaussian with mean $\mu 1_d$, where $\mu \in \mathbb{R}$ and $1_d$ is a $d$-dimensional vector of all ones. We vary $d \in \{1, 8, 64\}$ and

$$\mu \in \begin{cases} [-6, 6], & d = 1 \\ [-3, 3], & d = 8 \\ [-1, 1], & d = 64. \end{cases} \tag{23}$$

These ranges were selected to ensure that the total variation distance between the real and synthetic distributions at the extreme ends of the $\mu$ interval is at least 0.99, see Figure 1. We set the real and synthetic dataset sizes to 1000.

In the "with outlier" case, we add one outlier point to either the real or synthetic data at the largest $\mu$ value.

**Success Criteria**   As $\mu$ increases, the real and synthetic distribution are initially almost disjoint, then become more similar until they are identical at $\mu = 0$, and then become less similar until they are again almost disjoint at the right extreme. As such, to pass D1b (purpose), we require all metrics to have a bell-shaped curve centered at $\mu = 0$. For D4 (bounds), we require all metrics to be close to 0 at the left and right extremes, and close to 1 at $\mu = 0$.

The criteria for the "with outlier" case are the same, since a single outlier should not significantly change synthetic data evaluations.

## B.3. Gaussian Standard Deviation Difference

In this check, the real distribution is a $d$-dimensional Gaussian standard Gaussian distribution, and the synthetic distribution is a similar Gaussian with covariance $\sigma^2 I_d$, where $\sigma > 0$ and $I_d$ is the $d \times d$ identity matrix. We vary $d \in \{1, 8, 64\}$ and logarithmically vary $\sigma$ in the interval

$$\sigma \in \begin{cases} [10^{-3}, 10^3], & d = 1 \\ [10^{-1}, 10^1], & d = 8 \\ [10^{-0.5}, 10^{0.5}], & d = 64. \end{cases} \tag{24}$$

As in the Gaussian mean difference check, we selected the $\sigma$ intervals to ensure that the total variation distance between the real and synthetic distributions at the extreme ends is at least 0.99, see Figure 2. We set the real and synthetic dataset sizes to 1000.

**Success Criteria**   In this check, the synthetic distribution is very narrow with the smallest $\sigma$ values, but contained in the real distribution, so fidelity metrics should have high values and diversity metrics should have low values. As $\sigma$ increases,

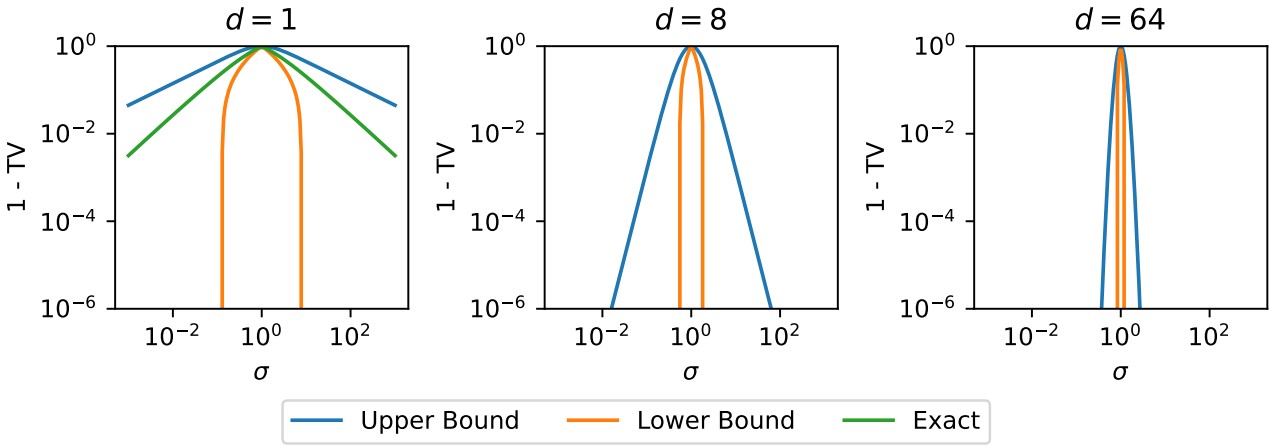

*Figure 2.* Total variation distance between a standard Gaussian and a Gaussian with covariance $\sigma^2 I_d$. The lower and upper bounds are based on Hellinger distance.

the metrics become more similar, and become identical at $\sigma = 1$, at which point all metrics should have high values. As $\sigma$ increases further, the synthetic distribution becomes wider, so fidelity metrics should drop. Diversity metrics can either fall to low value or stay at high values, as described in Section 4.2.

Our specific criteria are the following:

- For D1b (purpose):
    - Fidelity metrics should have a high-to-low shape.
    - Low metrics should have a bell-shape with midpoint $\sigma = 1$, high metrics should have a low-to-high shape.

- For D4 (bounds):
    - Fidelity metrics should have values close to 1, 1, 0 at the left extreme, $\sigma = 1$, and at the right extreme, respectively.
    - Diversity metrics should have values close to 0, 1 at the at the left extreme and $\sigma = 1$, respectively. Low metrics should have values close to 0 and high metrics should have values close to 1 at the right extreme.

**B.4. Sequential / Simultaneous Mode Dropping**

This checks evaluates how the metrics detect dropped modes. The real data is a 10-component Gaussian mixture in $d \in \{1, 8, 64\}$ dimensions, with the component means evenly spaced between 0 and $10 \cdot 1_d$, and the component standard deviations are

$$\sigma = \begin{cases} \frac{1}{6}, & d = 1 \\ \frac{1}{3}, & d = 8 \\ 1, & d = 64. \end{cases} \tag{25}$$

We set the real and synthetic dataset sizes to 1000.

We drop up to 9 of the modes in the synthetic distribution either simultaneously or sequentially, as done by Naeem et al. (2020). In simultaneous dropping, the weights of the 9 modes are gradually decreased, until their weight is zero and the synthetic distribution is just a single mode. In sequential dropping, we completely remove 0 to 9 of the modes from the synthetic distribution.

**Success Criteria** With both simultaneous and sequential dropping, the real and synthetic distributions are initially identical. Since the synthetic distribution is contained in the real distribution regardless of the dropped modes, fidelity metrics have high values in all cases. Diversity metrics should be sensitive to mode dropping, and decrease as mode modes are dropped. Our specific criteria are:

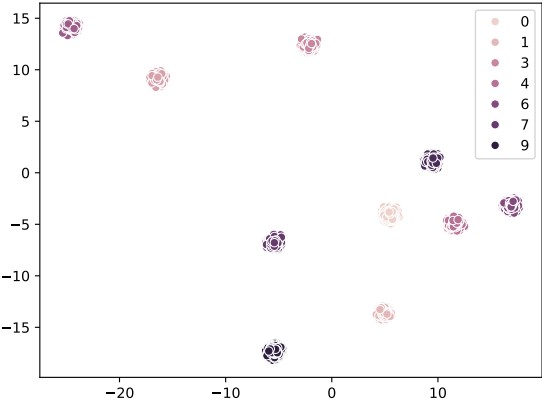

*Figure 3.* Samples of each component in the mode dropping and invention sanity check.

- For D1b (purpose):
    - Fidelity metrics should have horizontal lines.
    - Diversity metrics should have a high-to-low with middle drop shape, with a drop of at least 0.1, at the 0.95 quantile in the simultaneous case, and at the 0.5 quantile for the sequential case.

- For D4 (bounds):
    - Fidelity metrics should be close to one at both left and right extremes.
    - Diversity metrics should be close to one at the left extreme.

### B.5. Mode Dropping + Invention

This check from Kynkäänniemi et al. (2019) evaluates how the metrics detect both mode dropping and invention. We start be defining 10 two-dimensional Gaussian component distributions, which we show in Figure 3. The standard deviation of each component is 0.25, and the means were randomly sampled from a Gaussian with mean 0 and standard deviation 10. These same components are used for all repeats of the experiment. The real distribution is a mixture of components 0 to 4, and the synthetic distribution has between 1 and 10 components, included in order. As a result, the 5-component synthetic distribution is identical to the real distribution. With less than 5 modes, the synthetic distribution is dropping modes, and with more than 5 modes, it is inventing modes. We set the real and synthetic dataset sizes to 1000.

**Success Criteria**

- For D1b (purpose):

    - Fidelity metrics should have a high-to-low shape.
    - Diversity metrics should have a low-to-high shape.

- For D4 (bounds):

    - Both fidelity and diversity metrics should be close to 1 at 5 synthetic components.
    - Fidelity metrics should be close to 1 with 1 synthetic component.
    - Diversity metrics should be close to 1 with 10 synthetic components.

### B.6. Hypersphere Surface

This check from Khayatkhoei & Abdalmageed (2023) highlights a problematic behavior in some metrics. The real distribution is a uniform distribution on the surface of a $d$-dimensional hypersphere with radius 1, and the synthetic distribution is a similar uniform distribution on the surface of a hypersphere of radius $r \in [0.1, 1.9]$. We set the real and synthetic dataset sizes to 1000, and vary $d \in \{2, 8, 128\}$.

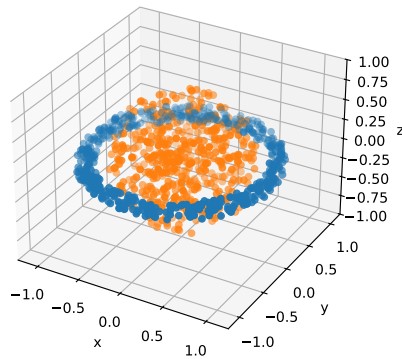

*Figure 4.* Samples from the sphere and torus distributions in the sphere vs. torus sanity check.

**Success Criteria**    With $r \neq 1$, the real and synthetic distributions are completely disjoint, so all metrics should have low values. In practice, detecting very small differences in $r$ is very hard for a general-purpose metric, so we only require that the metrics detect a difference at the extreme values of $r$. Specifically, for D1b (purpose), all metrics should have a bell shape with midpoint 1. For D4 (bounds), all metrics should have be close to zero at extreme values of $r$, and close to one at $r = 1$.

### B.7. Hypercube, Varying Sample / Synthetic Size

This check from Cheema & Urner (2023) evaluates how the metrics have with different sizes of real and synthetic data. The real distribution is uniform distribution on a $d$-dimensional hypercube with side length 1. The synthetic distribution is a similar uniform distribution, but the hypercube has been translated equally on all axes so that the $d$-dimensional volume of the overlapping area between the cubes is 0.2.

We vary either both the real and synthetic dataset size with both of them equal, or only the synthetic dataset size, in which case the real dataset size $|\Phi_r| = 1000$. The changing size is varied logarithmically in $[10^2, 10^4]$. We also vary $d \in \{1, 8, 64\}$.

**Success Criteria**    According to Cheema & Urner (2023), both fidelity and diversity metrics should have a value of 0.2, the volume of the overlap between the cubes, in this setup, since that is the value of a theoretical notion of precision and recall that can be computed in this case. We expect the metrics to get close to this value with the largest dataset size for D1b (purpose). We also expect the metric values to converge as the size increases (converging line shape), and require convergence by size 1000. This is counted under D2 (hyperparameters) when only the synthetic data size changes, and under D3 (data) when the sizes of both datasets change.

### B.8. Sphere vs. Torus

This check from Cheema & Urner (2023) evaluates how the metrics distinguish two disjoint distributions with a non-trivial geometry. One of the distributions is a uniform distribution on a sphere with radius 0.8. The other distribution is on a torus with major radius 1, minor radius 0.1, which surrounds the sphere, but is disjoint from it. We look at both making the sphere distribution the real distribution, and making the torus distribution real.

We sample the sphere distribution by sampling a cube containing the sphere, and rejecting samples outside the sphere. We sample the torus by first uniformly sampling an angle $\theta \in [0, 2\pi]$, and sampling a point uniformly on the circle that is formed by intersecting the torus with the $y = 0$ plane on the positive $x$ side. The point is then rotated by $\theta$ around the $z$ axis to obtain the final sample. Figure 4 shows samples from both distributions.

We set the real dataset size $|\Phi_r| = 1000$. We logarithmically vary the synthetic dataset size $|\Phi_g| \in [10^2, 10^4]$.

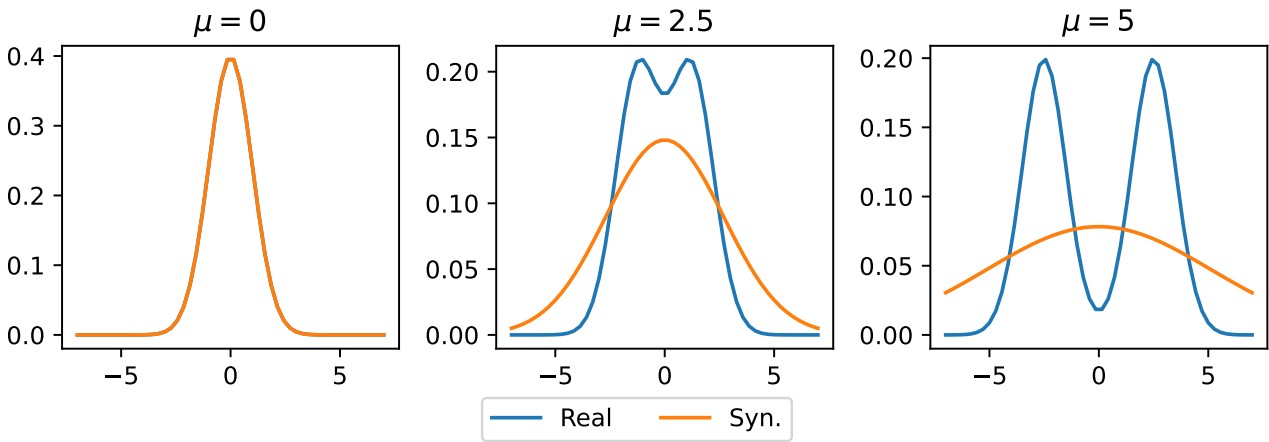

*Figure 5.* Comparison of pdfs for the real and synthetic distributions in the one vs. two modes check.

**Success Criteria** The two distributions in this check are disjoint, so all metrics should have low values. We do not place requirements at small synthetic dataset sizes, but we do require the metrics to pass regardless of which distribution is considered the real one. Specifically, for D1b (purpose), we require the metrics to have a converging line shape, with convergence at latest at 1000 synthetic datapoints. For D4 (bounds), we require the metrics to be close to 0 at the maximum number of synthetic datapoints.

### B.9. Mode Collapse

This check from Alaa et al. (2022) evaluates the mode resolution of the metrics: how well do they distinguish a real distribution of two modes from a synthetic distribution that models both modes with a single wide mode. The real distribution is an equal mixture of two $d$-dimensional ($d \in \{1, 8, 64\}$) Gaussians with means $-\frac{1}{2}\mu 1_d$ and $\frac{1}{2}\mu 1_d$ and unit variance, where $\mu \in [0, 5]$ is varied. The synthetic distribution is a Gaussian with mean 0 and covariance $(1 + \mu^2)I_d$. The size of both the real and synthetic dataset is 1000. See Figure 5 for a visualisation of the distributions with $d = 1$.

**Success Criteria** When $\mu = 0$, the distributions are identical. As $\mu$ increases, the modes of the real distribution become further separated, while the synthetic distribution widens to cover both modes. As a result, metric values should be high at low $\mu$ values, and decrease when $\mu$ is increased for fidelity metrics. Diversity metrics can either stay high or decrease depending on whether they are high or low metrics.

Our specific criteria are:

- For D1b (purpose):

  - Fidelity metrics should have a high-to-low shape.
  - Diversity metrics should have a horizontal line shape for high metrics, and a high-to-low shape for low metrics.

- For D4 (bounds): values should be close to one when $\mu = 0$ for all metrics.

### B.10. Scaling One Dimension

This check evaluates how robust the metrics are to differing scales of the data variables. The real data distribution is a two-dimensional standard Gaussian with the second dimension scaled by multiplying it with $s$. The synthetic data distribution is a similarly scaled Gaussian, but with mean $(6, 0)$. The scale is logarithmically varied in $s \in [10^{-3}, 10^3]$. The size of both datasets is 1000.

**Success Criteria** The two distributions are almost disjoint in the first dimension, so all metrics should have low values. Specifically, we require a horizontal line shape for D5 (invariance), and a values close to zero on both left and right extremes for desiderata D4 (bounds).

## B.11. Gaussian Mean Difference + Pareto

The aim of this check is to evaluate how well the metrics handle a heavy-tailed power-law distribution, which are common in tabular data. The setup is the same as in the Gaussian mean difference check with $d = 1$, but both the real and synthetic data contain an additional Pareto-distributed variable, which which has an identical distribution between the real and synthetic data. The specific Pareto distribution is type 1[9] with shape $\alpha = 1.01$ and scale 1.

**Success Criteria**    Since the presence of the Pareto-distributed variable with an identical distribution in the real and synthetic data should not affect data quality, the success criteria are the same as in the Gaussian mean difference check.

## B.12. One disjoint Dimension + Many Identical Dimensions

This check evaluates how the metrics handle a difference in just one of a large number of dimensions. The real distribution is a $d + 1$-dimensional standard Gaussian, and the synthetic distribution is a similar Gaussian, except the mean of the first dimension is 6. The number of identical dimensions $d$ is varied logarithmically in $d \in [1, 10^3]$. The sizes of the real and synthetic datasets are 1000.

**Success Criteria**    Since the two distributions are almost disjoint due to the difference in the first dimension, all metrics should have low values regardless of the number of extra dimensions. Specifically, for D1b (purpose), the values should be a horizontal line, and for D4 (bounds), they should be close to 0 at both left and right extremes.

## B.13. Discrete Numerical vs. Continuous Numerical

This check evaluates how the metrics detect a difference between a continuous distribution and a discrete numerical distribution. This difference can occur in both directions in tabular data. Many numerical values, such as a count of something, or age in years, are always integers in real data, but a synthetic data generator based on transforming a continuous distribution, like a GAN or a diffusion model, will likely model them as a continuous value.[10] In the other direction, some synthetic data generators (McKenna et al., 2022) must discretise continuous variables, and can only output the discretised values. Evaluation metrics should detect differences in both directions.

One distribution is a standard Gaussian multiplied by a scale $s$, and the other distribution is the same Gaussian and multiplier, but its values have been rounded to integers. We look at both the case where the continuous distribution is real, and the case where the discrete distribution is real. The scale $s$ is varied logarithmically in $s \in [1, 10^3]$. The sizes of the real and synthetic datasets are 1000.

**Success Criteria**    The two distributions in this check are almost completely disjoint, since one of them is continuous and one of them is discrete. However, when the discrete distribution is real, the continuous synthetic distribution covers all of its values, so high-type diversity metrics can have high values in this case. Conversely, when the continuous distribution is real, all values from the discrete synthetic distribution have reasonable density in the real distribution, so fidelity metrics should have high values.

Our specific criteria are:

- For desiderata D1b (purpose), all metrics should be horizontal lines regardless of which distribution is considered real.

- For D4 (bounds):

    - If the discrete distribution is real:
        * Fidelity metrics should be close to 0 at small and large discretisation scales.
        * High diversity metrics should be close to 1 and low diversity metrics should be close to 0 at small and large discretisation scales.
    - If the continuous distribution is real:
        * Fidelity metrics should be close to 1 at small and large discretisation scales.
        * Diversity metrics should be close to 0 at small and large discretisation scales.

---

[9]The pdf is $p(x) = \frac{\alpha}{x^{\alpha+1}}$ on $x \geq 1$.

[10]The continuous values can of course be rounded to integers, but evaluation metrics should detect a lack of rounding.

## C. Sanity Check Result Plots

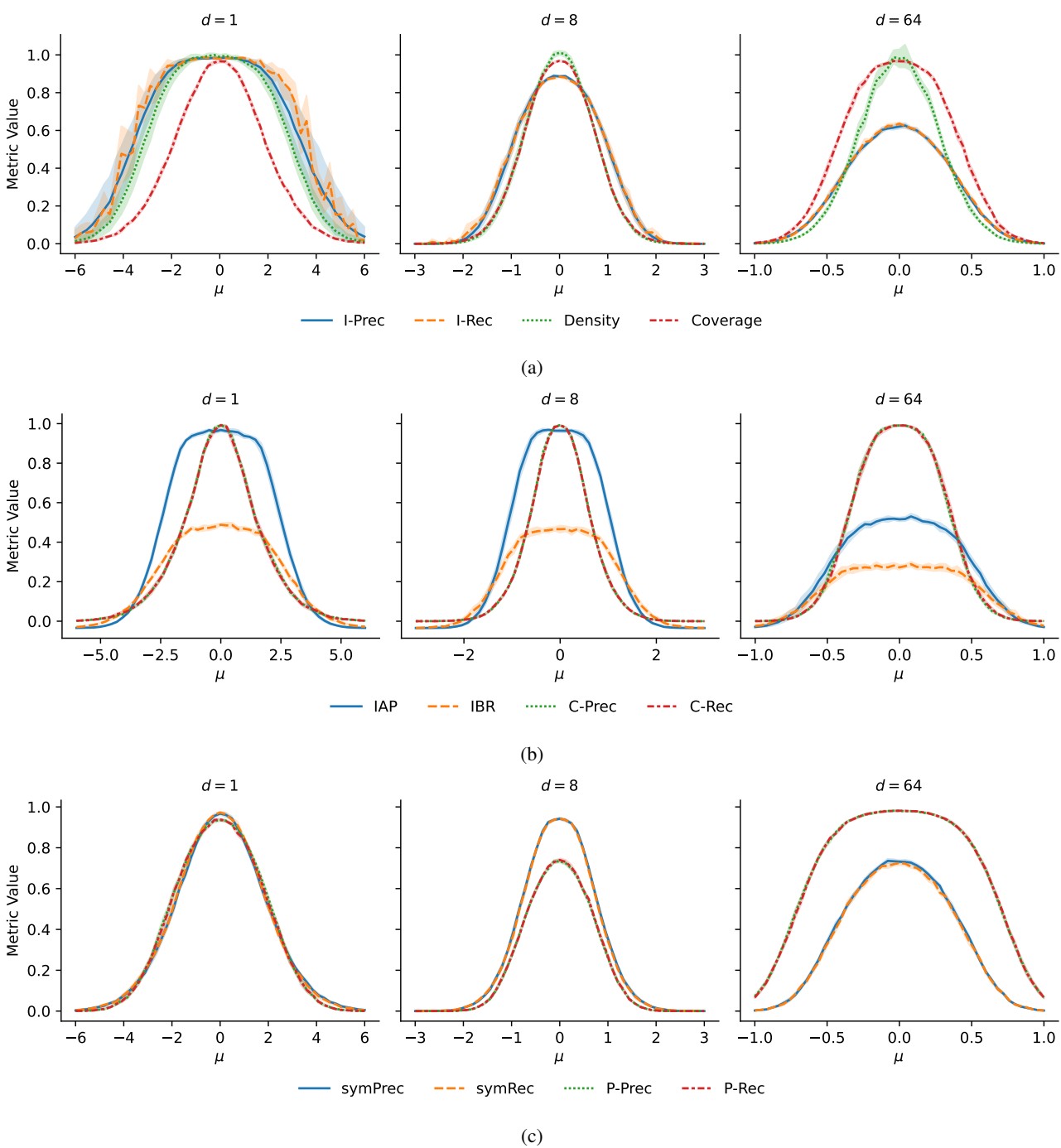

*Figure 6.* Gaussian mean difference check: two $d$-dimensional Gaussian distributions with real mean 0, synthetic mean $\mu$, covariance $I_d$.

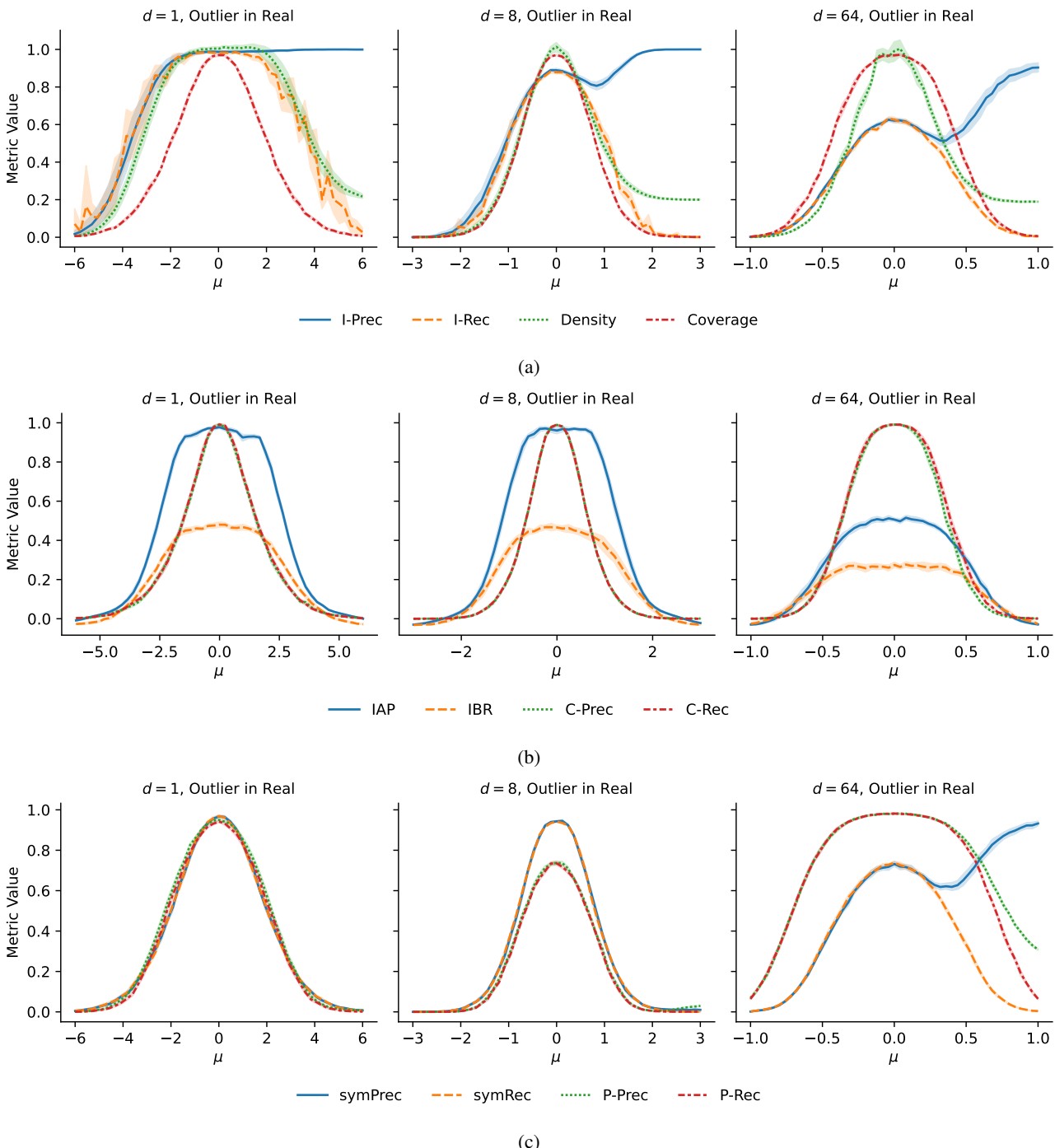

*Figure 7.* Gaussian mean difference + outlier (in real data) check: two $d$-dimensional Gaussian distributions with real mean 0, synthetic mean $\mu$, covariance $I_d$, with an outlier at the largest $\mu$ displayed on the x-axis in the real data.

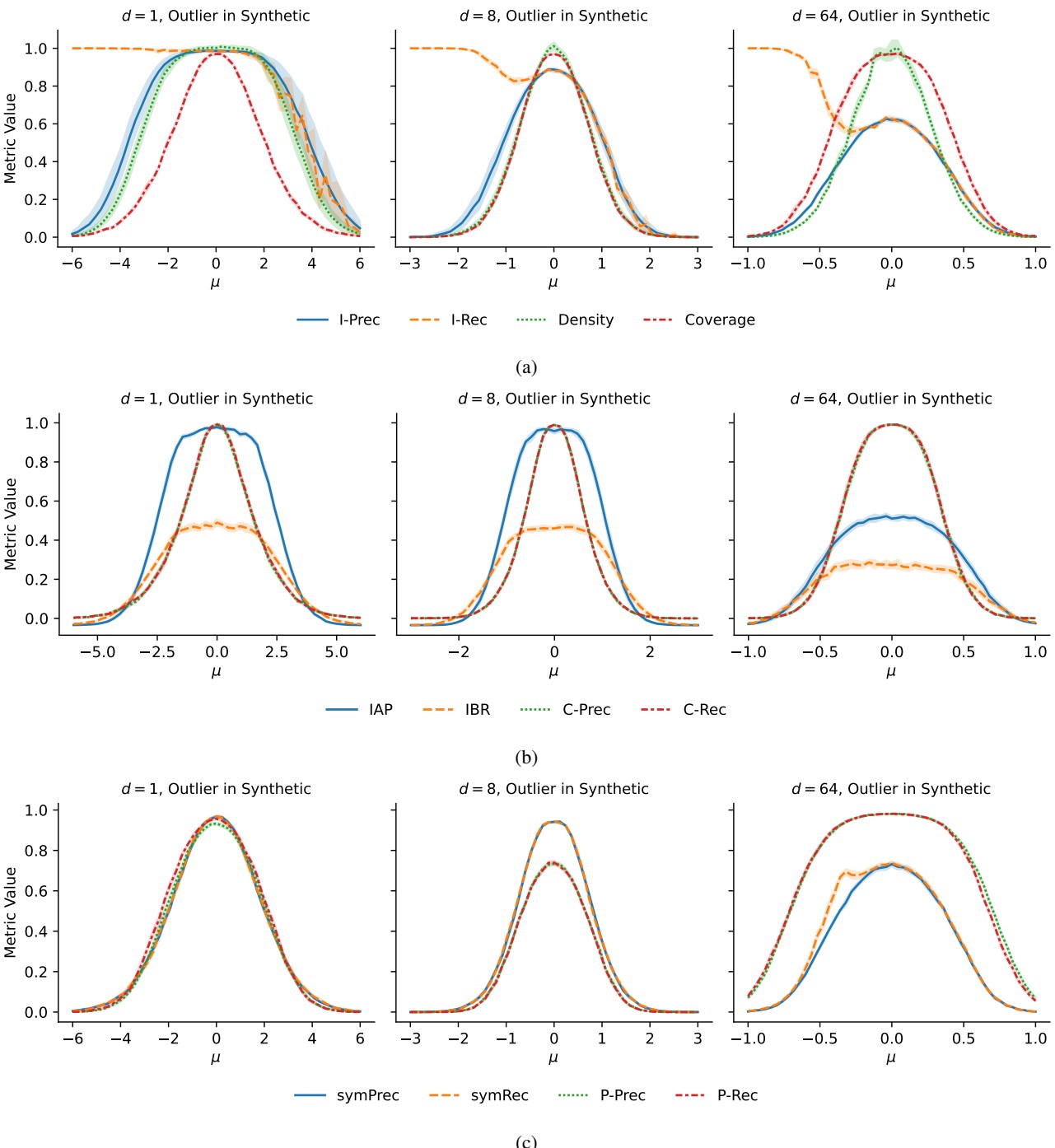

*Figure 8.* Gaussian mean difference + outlier (in synthetic data) check: two $d$-dimensional Gaussian distributions with real mean 0, synthetic mean $\mu$, covariance $I_d$, with an outlier at the largest $\mu$ displayed on the x-axis in the synthetic data.

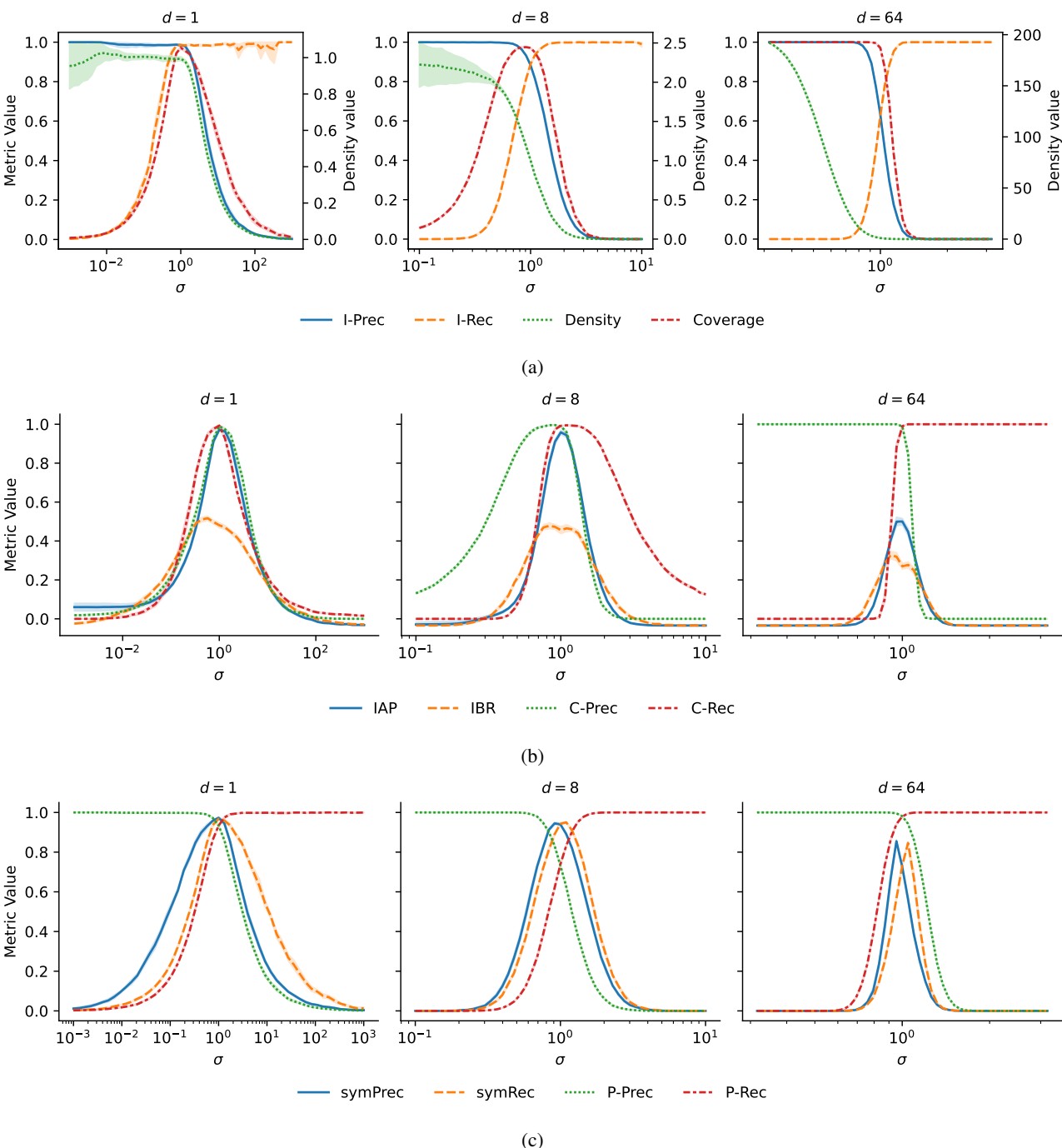

*Figure 9.* Gaussian standard deviation difference check: two $d$-dimensional Gaussian with mean 0, real standard deviation 1, and differing synthetic standard deviation. Density is plotted on a separate y-axis due to having a very different range from the other metrics.

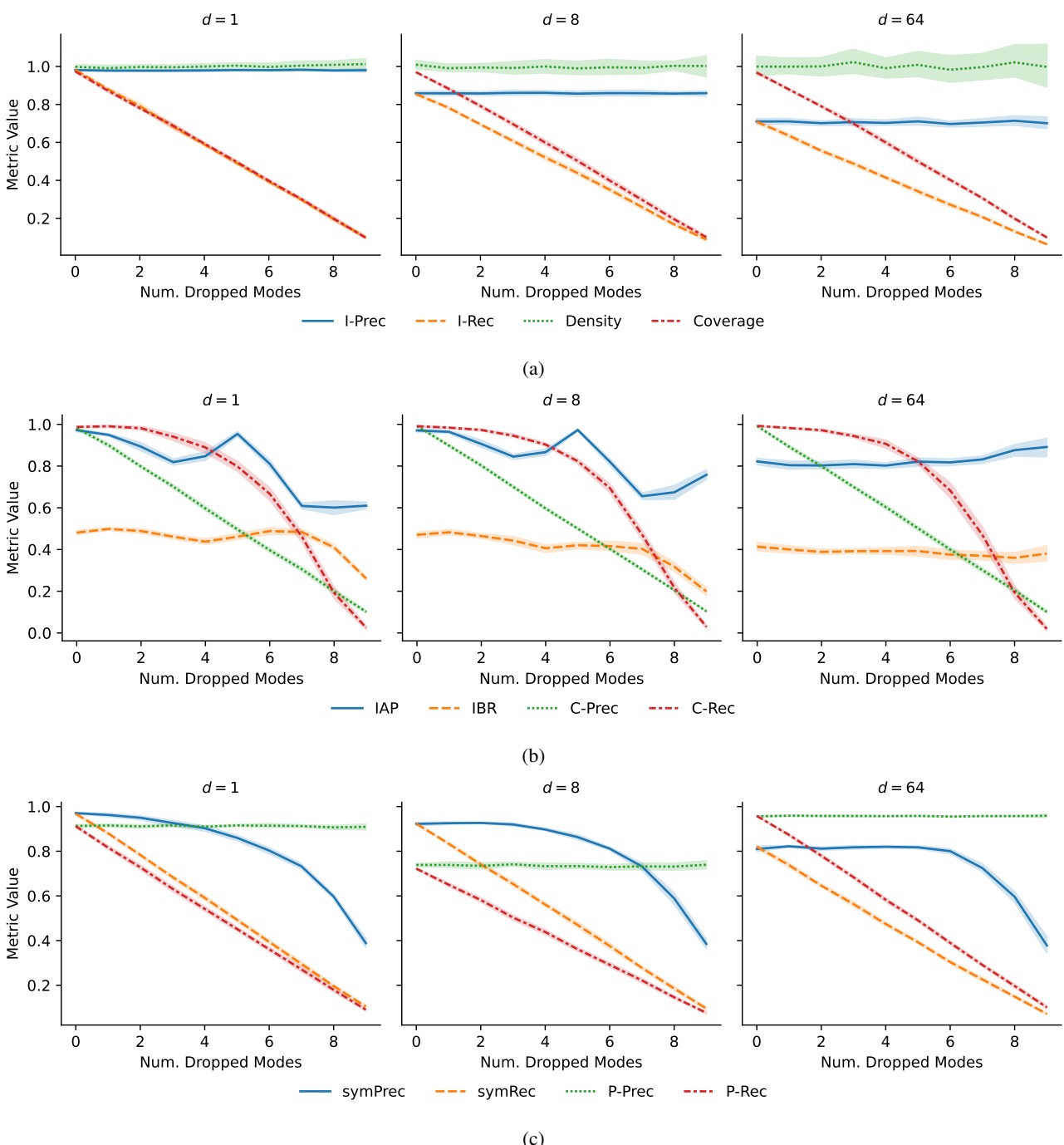

*Figure 10.* Sequential mode dropping check: two $d$-dimensional mixtures of Gaussians, where the real data has 10 modes, and the synthetic data drops 0 to 9 of these modes. Fidelity metrics should not drop, since the synthetic is always from a part of the real data distribution, but diversity metrics should drop due to the dropped modes.

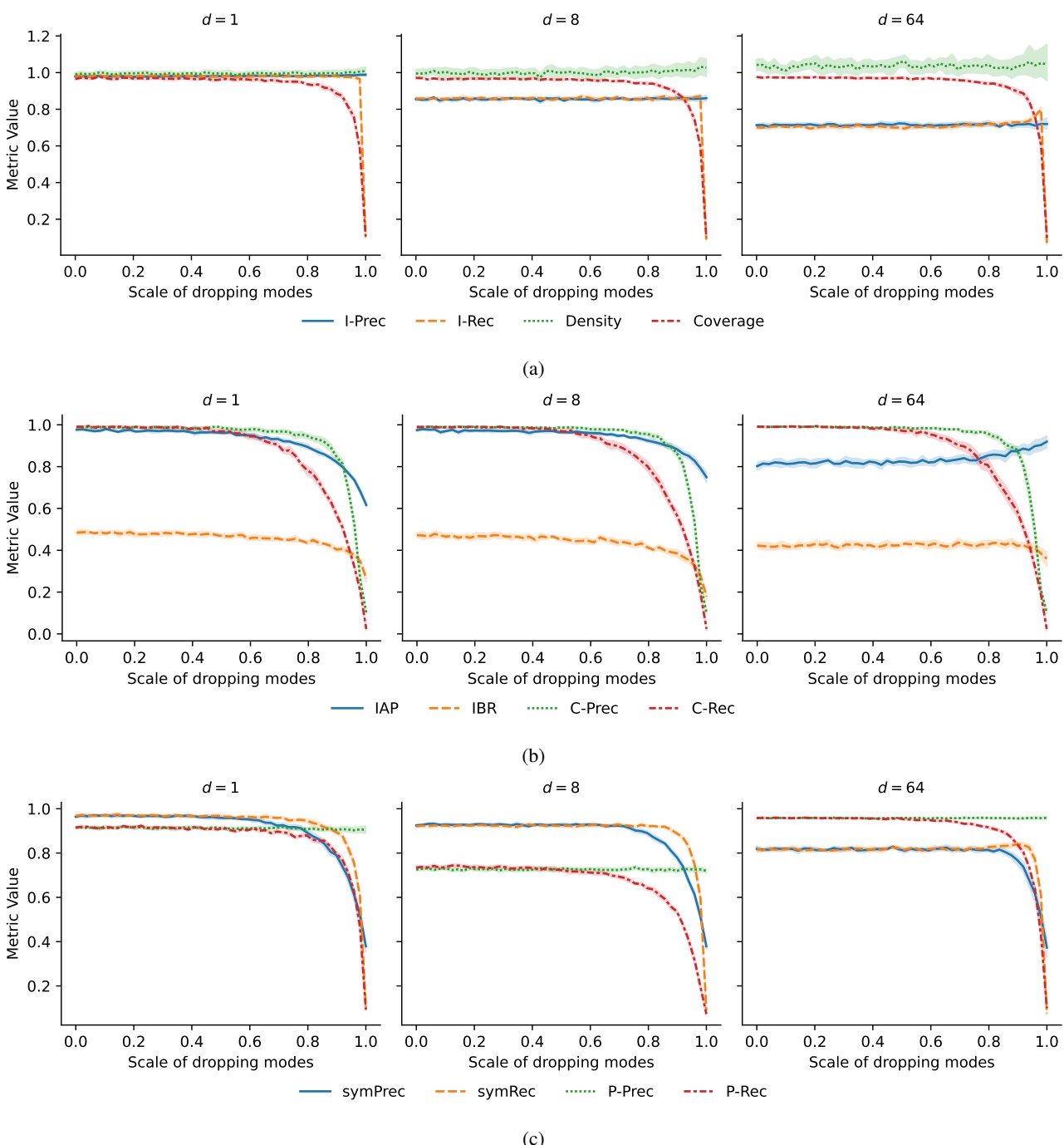

*Figure 11.* Simultaneous mode dropping check: two $d$-dimensional mixtures of Gaussians, where the real data has 10 modes, and the synthetic data scales down the density of 9 modes by the number on the x-axis. Fidelity metrics should not drop, since the synthetic is always from a part of the real data distribution, but diversity metrics should drop due to the dropped modes.

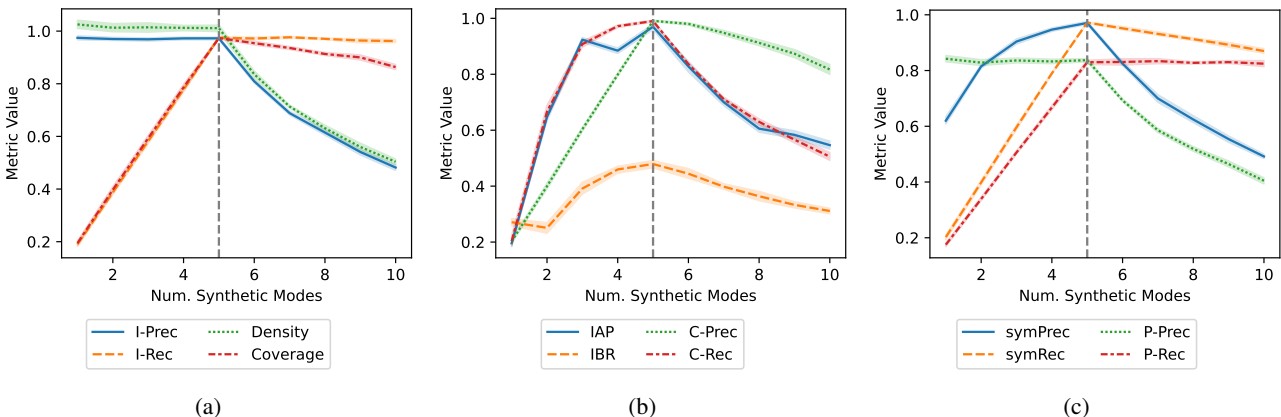

*Figure 12.* Mode dropping + invention check: 2-dimensional mixtures of Gaussians, where the real data has 5 modes, and the synthetic data has 1-10 modes. The first 5 are the same as the real data, and the last 5 are invented modes. Fidelity metrics should be close to 1 until 5 modes, and then drop, while diversity metrics should increase with 1 to 5 modes, and be close to 1 with 5 to 10 modes.

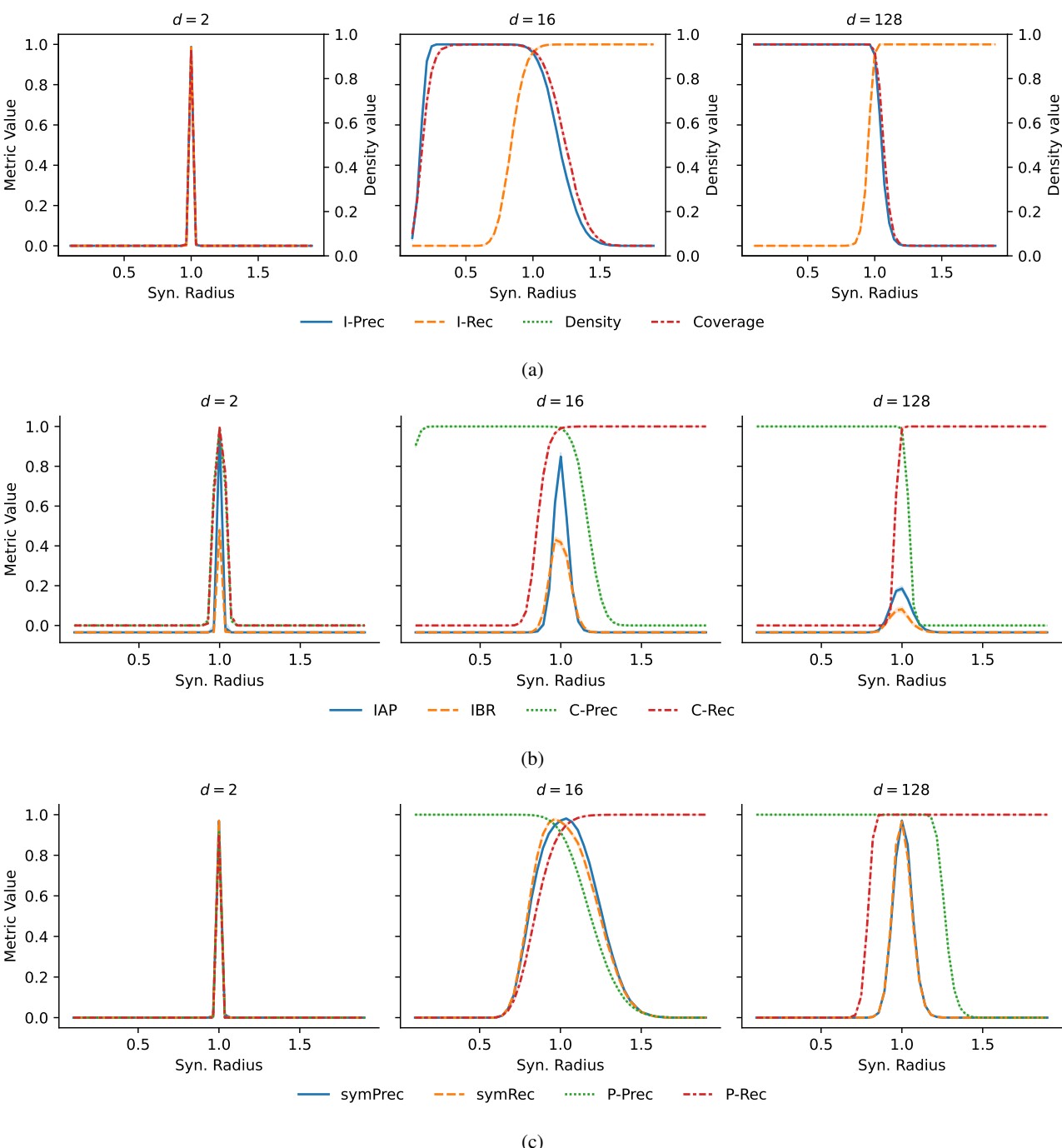

*Figure 13.* Hypersphere surface check: two uniform distributions on the surfaces of two hyperspheres, with radius 1 for real data and the radius on the x-axis for the synthetic data. When the synthetic radius is not 1, the distributions are completely disjoint, so all metrics should have the value 0 away from synthetic radius 1.

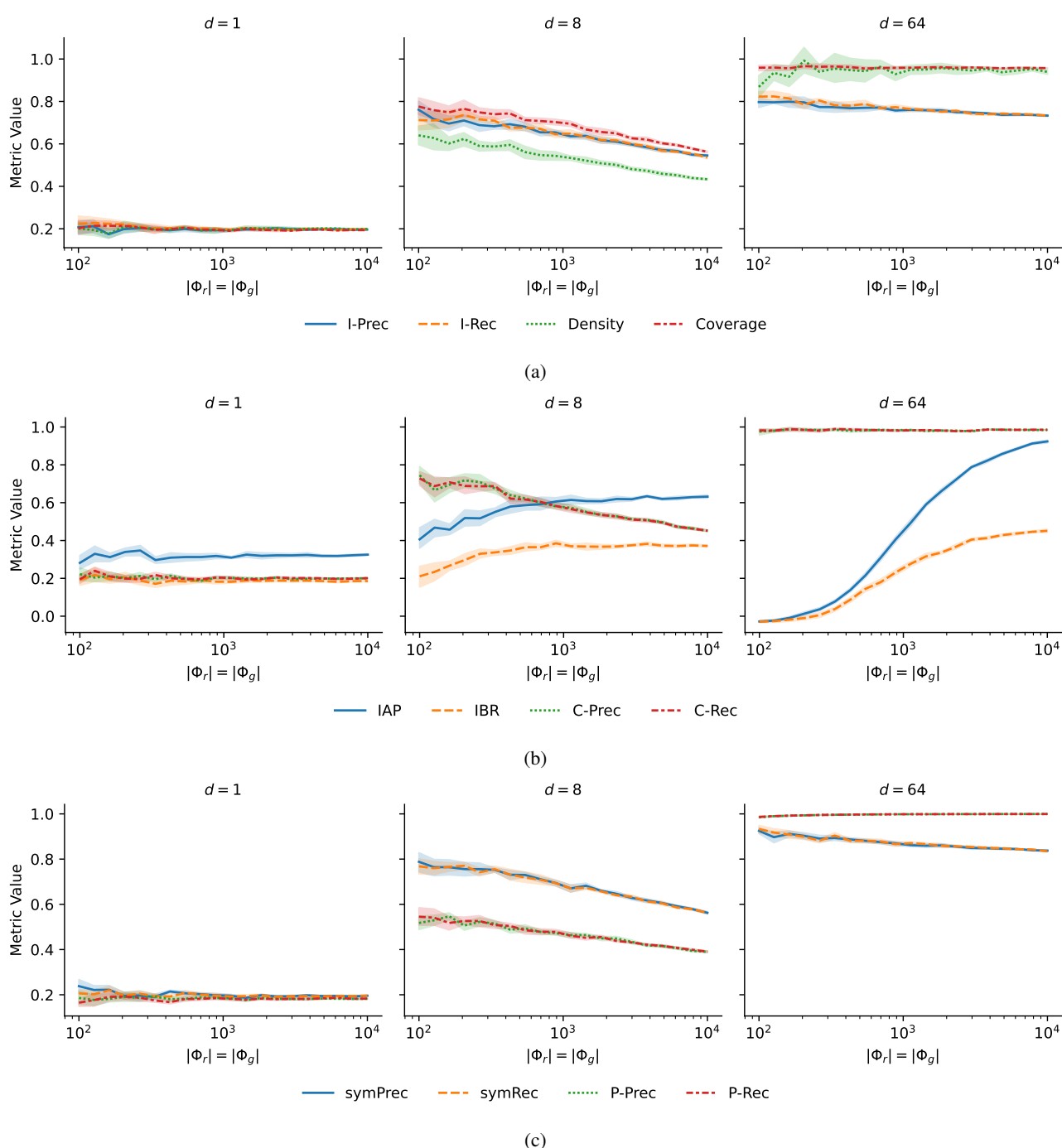

*Figure 14.* Hypercube, varying sample size check: two uniform distributions on overlapping hypercubes in $d$ dimensions, varying both the synthetic dataset size $|\Phi_g|$ and the real dataset size $|\Phi_r|$.

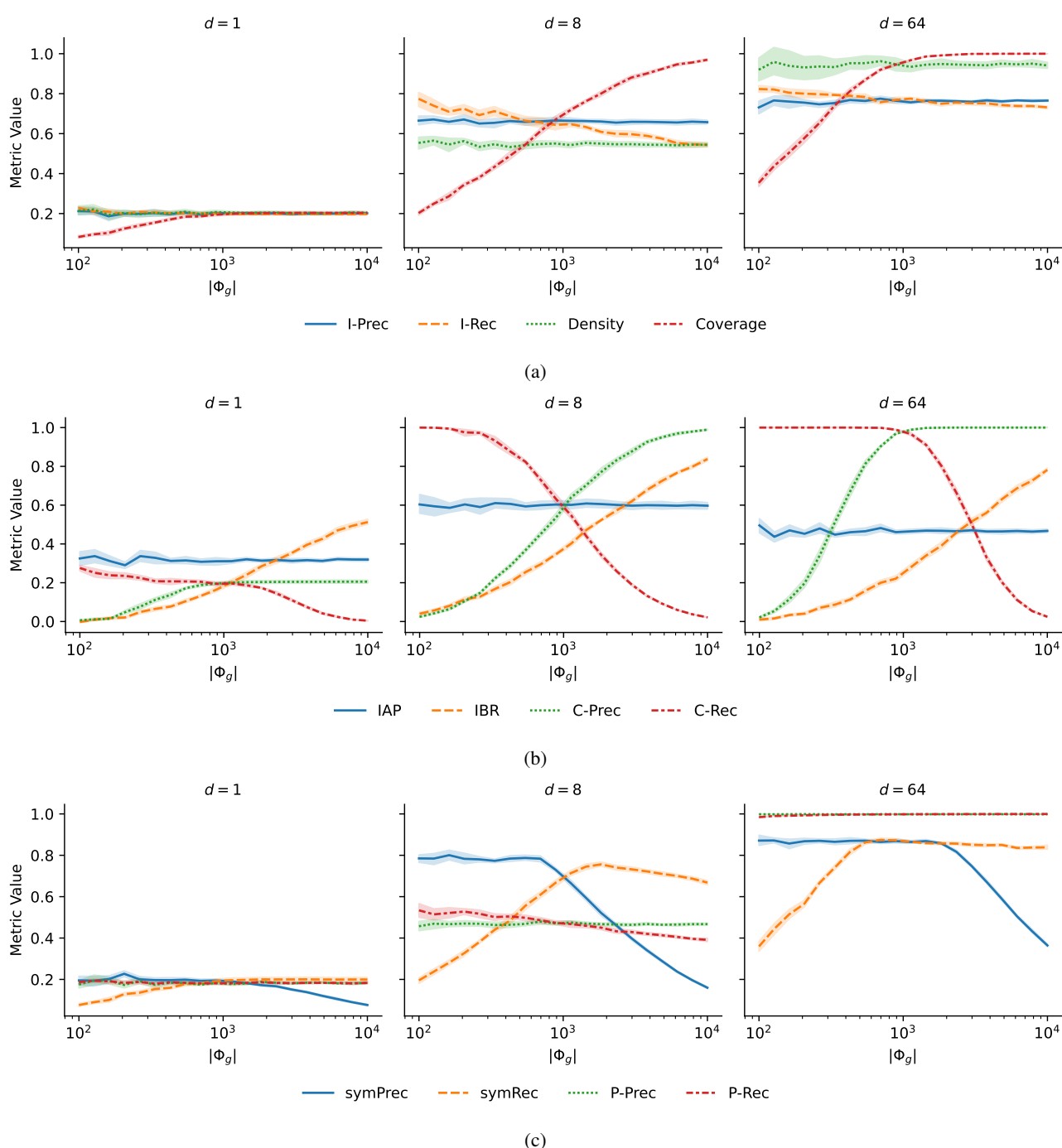

*Figure 15.* Hypercube, varying synthetic size check: two uniform distributions on overlapping hypercubes in $d$-dimensions, varying the synthetic dataset size $|\Phi_g|$, while the real dataset size if fixed at $|\Phi_r| = 1000$.

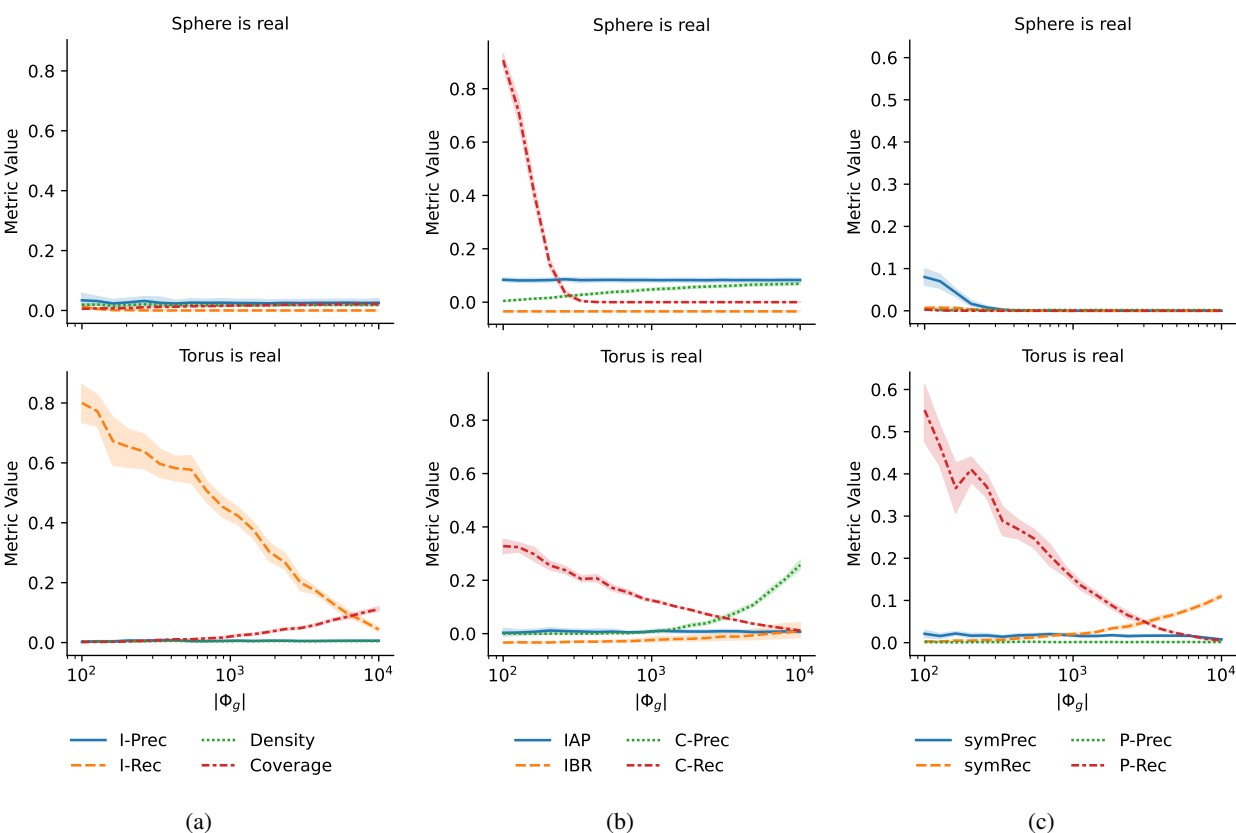

*Figure 16.* Sphere vs. torus check: distributions on a sphere and a disjoint torus surrounding the sphere, with varying synthetic dataset size. The real dataset size is fixed at $|\Phi_r| = 1000$.

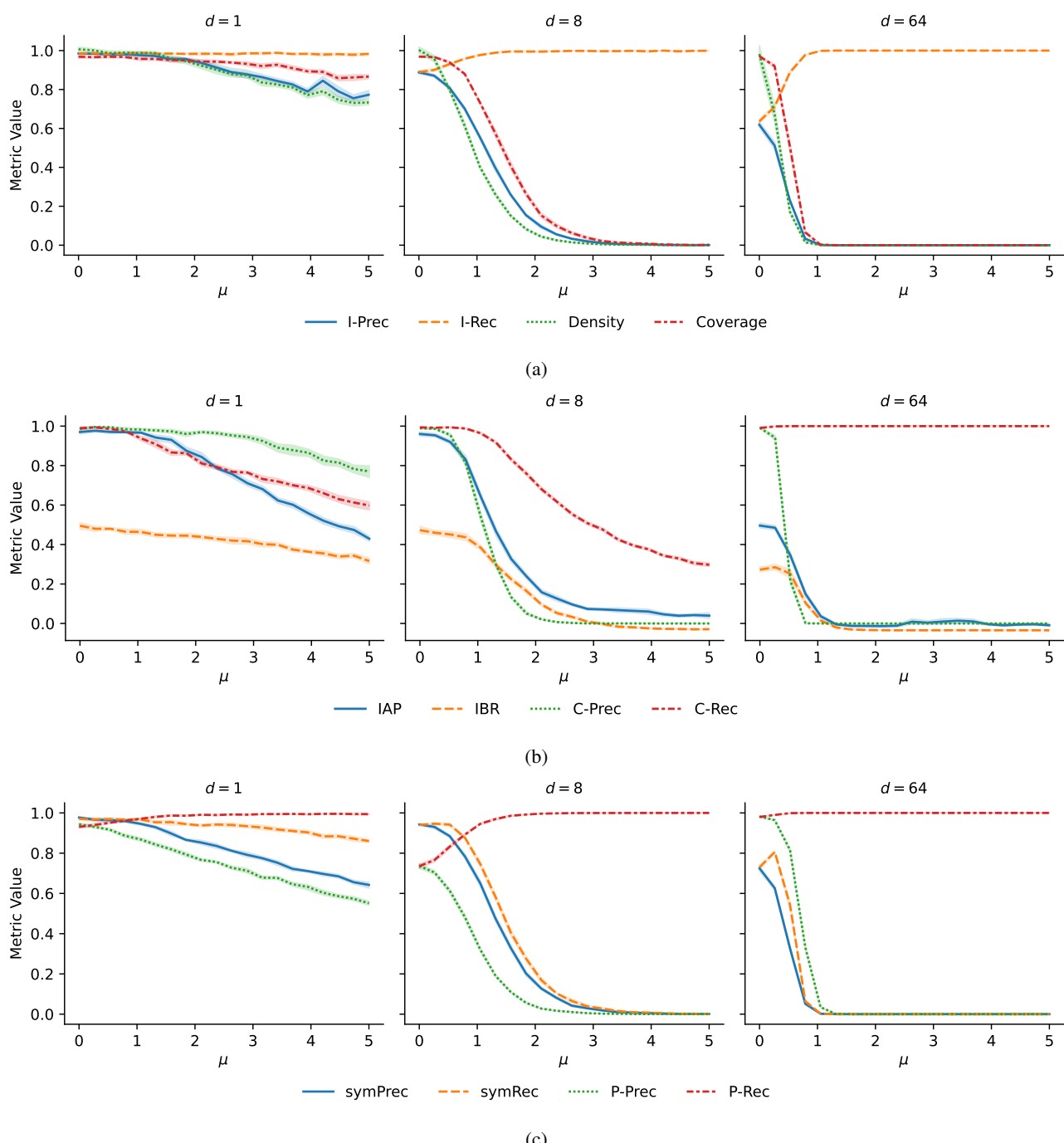

*Figure 17.* Mode collapse check: the real distribution being a two-component Gaussian mixture, and the synthetic distribution is a Gaussian that covers the real distribution. $\mu$ is the separation between the two components of the real distribution.

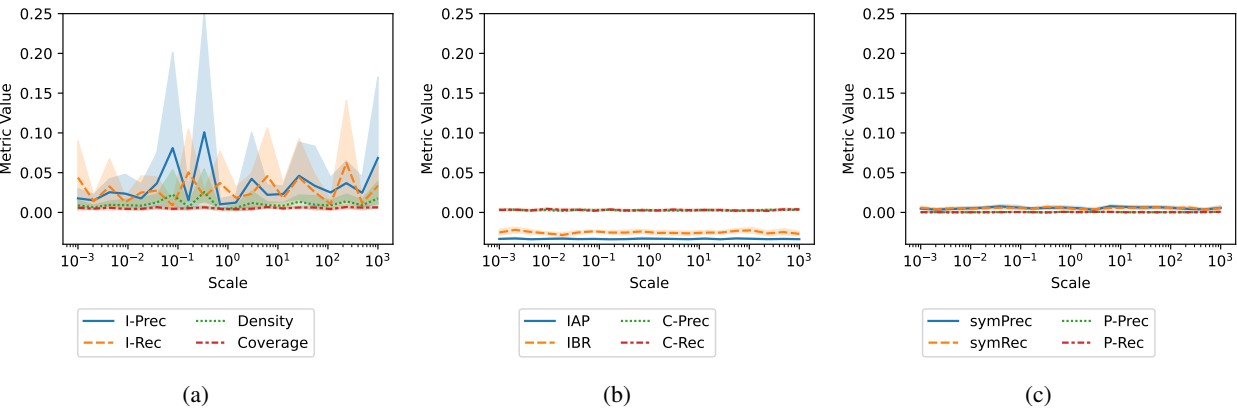

*Figure 18.* Scaling one dimension check: two 2-dimensional Gaussian distributions that are almost disjoint on the first dimension, identical on the second dimension, and the second dimension is scaled by a varying multiplier. Negative values of IAP and IBR are due to numerical integration errors.

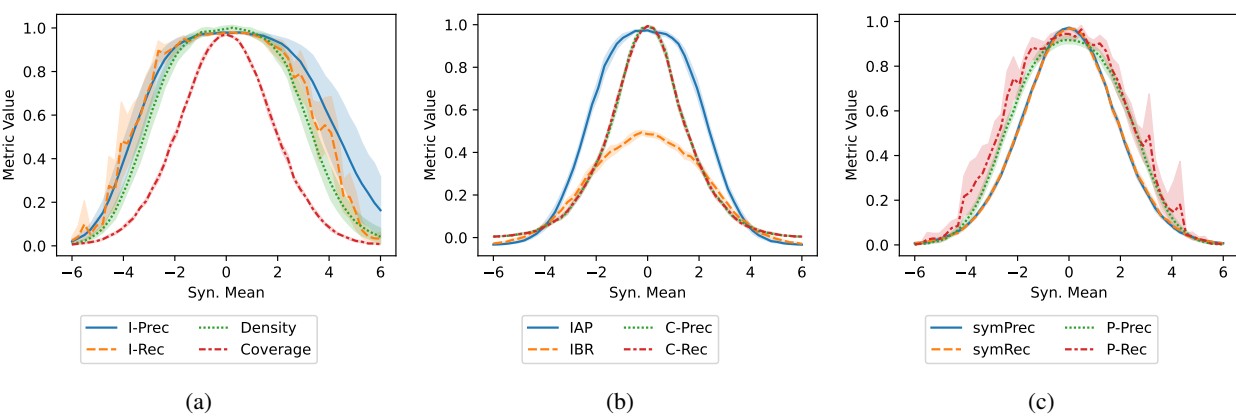

*Figure 19.* Gaussian mean difference + Pareto check: two-dimensional data, where the first dimension is Gaussian with mean different means in the real and synthetic data, and the second dimension is a Pareto distribution that is identical between real and synthetic data.

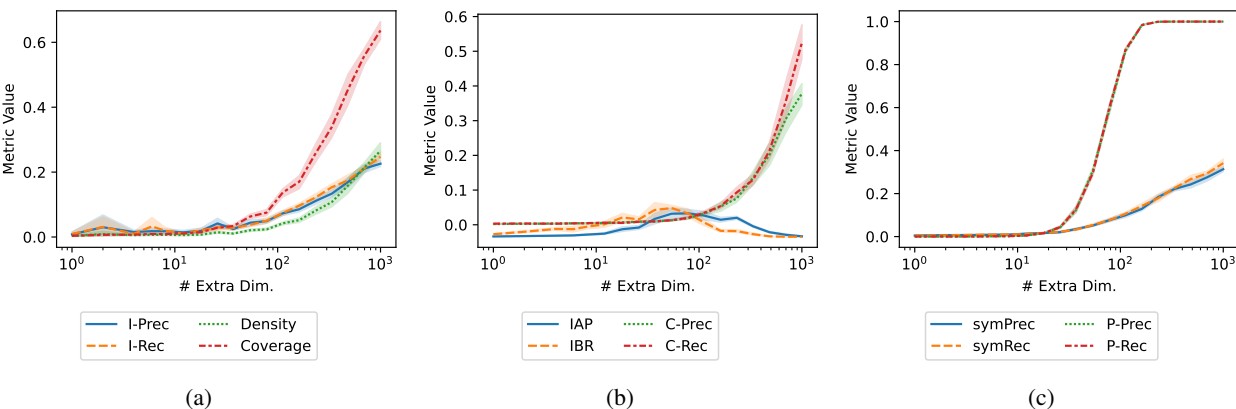

*Figure 20.* One disjoint dimension + many identical dimensions check: two Gaussian distributions that are almost disjoint in the first dimensions and identical in the rest of the dimensions, with varying number of the identical extra dimensions.

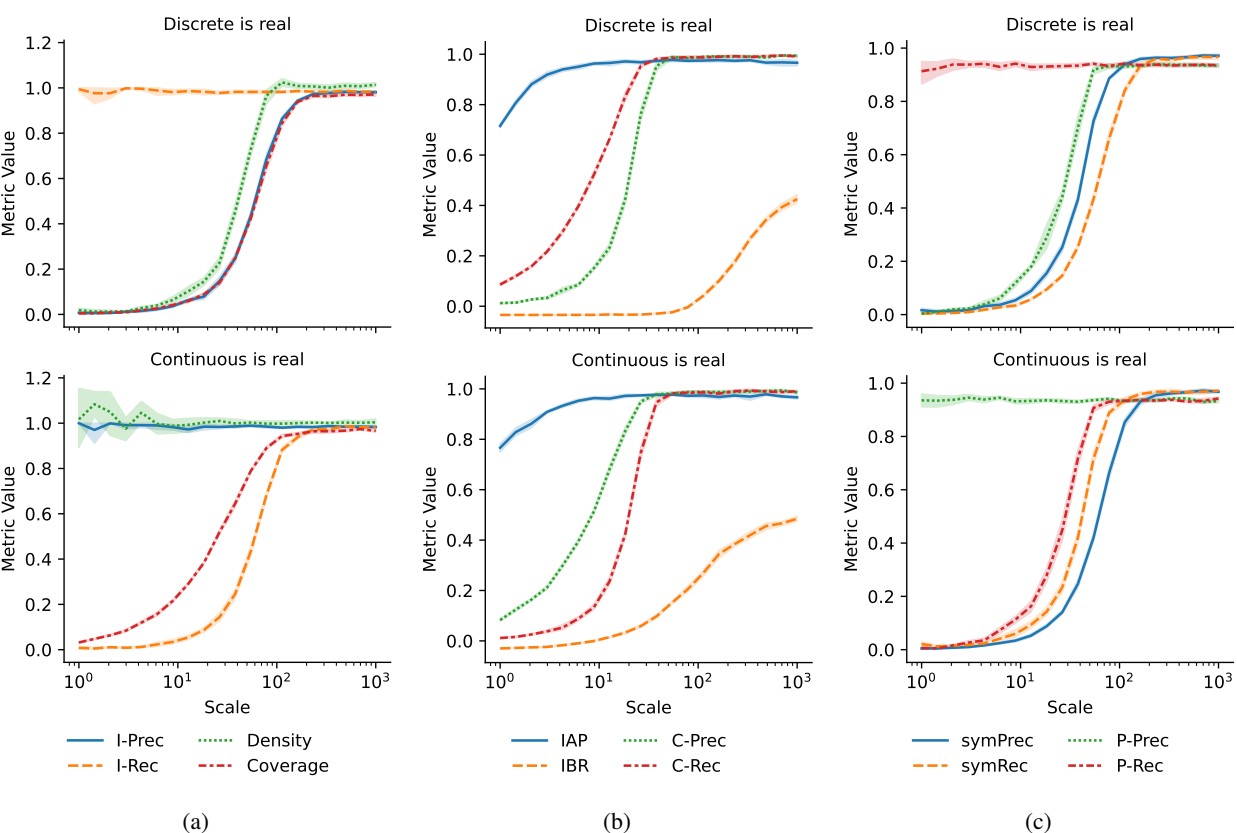

*Figure 21.* Discrete numerical vs. continuous numerical check: two identical scaled Gaussian distributions, one of which is discretised by rounding to integers.

