# OpenReview forum: "Position: All Current Generative Fidelity and Diversity Metrics are Flawed"
_ICML.cc/2025/Position_Paper_Track — ICML 2025 Position Paper Track poster_

### Official Review · Reviewer_QPmc · 2025-03-03

**Significance:** 2
**Argument Clarity:** 3
**Rating:** 3
**Confidence:** 4

**Questions:**

Pls see. "Weaknesses" 2,4.

**Discussion Potential:**

3

**Paper Summary:**

The paper states that "All Current Generative Fidelity and Diversity Metrics are Flawed". Previous works have found many failure cases in current metrics, for example lack of outlier robustness and unclear lower and upper bounds. A comprehensive list desiderata for such measures and experiments with synthetic datasets is provided. After doing experiments, authors arrive at a position: all current generative fidelity and diversity metrics are flawed, because each metric doesn't pass some test.

**Position:**

No

**Position In Title:**

Yes

**Related Work:**

3

**Strengths And Weaknesses:**

Strengths:
1. To the best of my knowledge, this is the first attempt to provide a large-scale empirical evaluation of generative models' metrics.
2. A lot of experiments with synthetic datasets are provided.

Weaknesses:
1. **The main weakness is that the paper is de facto an original research, not a "position" which should give a review of existing papers and show a promising research direction, call to action, a value statement, a policy proposal, etc.**
Only this point is a valid reason for rejection. The rest of my comments are about contents of the paper, as if it were submitted to the research track.

2. Some experiments are hard to understand, i.e. why "Hypercube, varying sample size check" is failed for all the metrics? Very few details are provided to the setting of the experiment and its interpretation.
3. Relevant citations are missing:

Barannikov, S., Trofimov, I., Sotnikov, G., Trimbach, E., Korotin, A., Filippov, A., & Burnaev, E. (2021). Manifold Topology Divergence: a Framework for Comparing Data Manifolds. Advances in neural information processing systems, 34, 7294-7305.

Kim, P. J., Jang, Y., Kim, J., & Yoo, J. (2023). Topp&r: Robust support estimation approach for evaluating fidelity and diversity in generative models. Advances in Neural Information Processing Systems, 36, 7831-7866.

Southern, J., Wayland, J., Bronstein, M., & Rieck, B. (2023). Curvature filtrations for graph generative model evaluation. Advances in Neural Information Processing Systems, 36, 63036-63061.

Borji, A. (2022). Pros and cons of GAN evaluation measures: New developments. Computer Vision and Image Understanding, 215, 103329.

4. Why Section 2 is called "Desiderata for Tabular Synthetic Data Evaluation Metrics"? Refereces to tabular data here and in Table 3 are not grounded.

5. "However, many works evaluate a correlation between a metric and something that is as easy to compute as the metric, like the accuracy of predictive models trained on synthetic data. We
do not think that these evaluations are meaningful, since a practitioner could compute the quantity they care about..."

I disagree with this point, because training on synthetic data can take a lot of time and be not practical. For example, one can't do it every epoch for early stopping.

6. Many checks are copied from previous research like Uniform Hypersphere Surface Check, Torus vs. Sphere Check; this fact reduces the novelelty.

7. The paper is poorly organized, for example the check "Discrete Numerical vs. Continuous Numerical" is not defined in the main text.

8. No experiments with real data is provided. For example, mode drop/invention can be easily checked with datasets like ImageNet, CIFAR, etc.

9. Simple FID is not included.

**Support:**

3

---

> ### Author Rebuttal · Authors · 2025-03-31
>
> Thank you for the review.
>
> > The main weakness is that the paper is de facto an original research, not a "position" which should give a review of existing papers and show a promising research direction, call to action, a value statement, a policy proposal, etc.
>
> Our paper does have original research, but it fits the position paper track well. First, our paper is about what *should* be done, exactly what the call for position papers asks for in the first paragraph. From the more specific list in the call you have partially quoted, our paper contains a proposed research direction (better metrics), and two recommendations on how research should be conducted (generative model evaluations should be wary of flaws in metrics and evaluation of metrics should contain a broad set of sanity checks). Second, there is nothing in the call saying that position papers cannot contain original research to support their position.
>
> We include the original research, namely the benchmark of the metrics on the sanity checks, as a justification and evidence to support our position. Had we not implemented this benchmark and instead only relied on existing work, our position would have become much weaker, since in that case, we could only have speculated that metrics might fail checks that they haven’t been evaluated on. By running the sanity checks, we now know which metrics fail which checks.
>
> > Some experiments are hard to understand, i.e. why "Hypercube, varying sample size check" is failed for all the metrics? Very few details are provided to the setting of the experiment and its interpretation
>
> We have included many of the details in the Appendix. Appendix C contains detailed descriptions of the experiments and the success criteria, and Appendix A.2 discusses the hardest sanity checks, including "Hypercube, varying sample size". We will add more details to the main text in the revision.
>
> > Relevant citations are missing:
>
> Thank you for pointing these out, we will include them in the revision. We attempted to evaluate the metrics from Kim et al. (2023) during the rebuttal, but their implementation was not able to compute even a single evaluation in reasonable time.
>
> > Why Section 2 is called "Desiderata for Tabular Synthetic Data Evaluation Metrics"? Refereces to tabular data here and in Table 3 are not grounded.
>
> The word “tabular” in the title is simply a leftover from an earlier draft, we will remove it in the revision.
>
> > "However, many works evaluate a correlation between a metric and something that is as easy to compute as the metric, like the accuracy of predictive models trained on synthetic data. We do not think that these evaluations are meaningful, since a practitioner could compute the quantity they care about..."
>
> > I disagree with this point, because training on synthetic data can take a lot of time and be not practical. For example, one can't do it every epoch for early stopping.
>
> This is a good point that we have not considered. We will remove the statement that previous works have done unmeaningful evaluations with downstream predictive accuracy as the target metric. This fixes both this issue, and also improves the clarity of the paragraph that reviewer TDnJ asked for.
>
> > Many checks are copied from previous research like Uniform Hypersphere Surface Check, Torus vs. Sphere Check; this fact reduces the novelelty.
>
> Yes, the majority of our checks are from the literature. We have included them, since their purpose in the literature was to point out deficiencies of other metrics. Our contribution with these checks is evaluating all of them together, not only a narrow set of checks each of the previous works did.
>
> > The paper is poorly organized, for example the check "Discrete Numerical vs. Continuous Numerical" is not defined in the main text.
>
> This check is introduced in Section 4.1 under the paragraph “Tabular Data Focused Checks”. We will clarify this paragraph to make the association clearer.
>
> > No experiments with real data is provided. For example, mode drop/invention can be easily checked with datasets like ImageNet, CIFAR, etc.
>
> We give our reasons to not use real data in Section 5.3. To summarise, with real data, we would not be able to focus on a single potential flaw as easily as with simple toy data, and we would not know the values the metrics should have precisely enough to evaluate D4 (bounds). For example, consider a generator that generates only images of cats when the real data only has images of trucks. It is clear that this generator should get low fidelity and diversity scores, but it is not clear that these scores should be 0, since images of cats and trucks can still have some commonalities. For example, an image of a cat could have a truck in the background.
>
> > Simple FID is not included.
>
> We focus on fidelity and diversity metrics, as explained in the Introduction. FID is not in either group, so it is not included.

---

> > ### Comment · Reviewer_QPmc · 2025-04-04
> >
> > Thank you for a detailed answer. After a careful reading of ICML rules I came to a conlcusion that position track papers are not forbided to include experimental results. To be precise, they shouldn't describe new research without advocating a position. I agree with you at this point, my critique was not justified.
> > Most of my questions are addressed. However, some remain.
> > 1) > Discrete Numerical vs. Continuous Numerical
> >
> > I think that this is not a good idea to include such sanity check, because it is related only to tabular data.
> > For the more common image data, one can study distributions of real and generated data and compare them geometrically via Precision/Recall. At the same time, "Discrete Numerical vs. Continuous Numerical" is a completely different kind of a test specific to one-dimensional data.
> >
> > 2) After an evaluation of several popular metrics can you conclude which is the best one for now?

---

> > > ### Author Response · Authors · 2025-04-05
> > >
> > > Thank you for reconsidering your review and previous evaluation.
> > >
> > > > Discrete Numerical vs. Continuous Numerical
> > >
> > > > I think that this is not a good idea to include such sanity check, because it is related only to tabular data. For the more common image data, one can study distributions of real and generated data and compare them geometrically via Precision/Recall. At the same time, "Discrete Numerical vs. Continuous Numerical" is a completely different kind of a test specific to one-dimensional data.
> > >
> > > We agree that this check is not relevant to image data, or other domains outside tabular data. We also recognize that including the check makes it harder for readers outside tabular data to see which checks are relevant for them. However, if we removed the check, readers working on tabular data would not learn any insight that the check provides. The majority of real-world data is tabular (Shwartz-Ziv and Armon 2022), and these metrics are used to evaluate tabular generative models, for example in Kotelnikov et al. (2023) and Zhang et al. (2023). As a result, we think that the harm to tabular data readers from not including the check (no information from the check) far outweighs the harm to non-tabular data readers from including the check (inconvenience from needing to ignore the check), so we will keep the check. To mitigate the resulting inconvenience to non-tabular data readers, we will explicitly mark checks relevant to only tabular data in Tables 2 to 4 in the revision.
> > >
> > > > After an evaluation of several popular metrics can you conclude which is the best one for now?
> > >
> > > We cannot conclude that there is a single best metric for all settings, since every metric fails some check(s) that others pass. In other words, there is no magic bullet: using one (or just a few) tests will mean that there are inevitably potential failure modes that will go unnoticed. We have given recommendations on which metrics to use, depending on the setting, in Appendix A.1, so our readers can select metrics for their specific needs.
> > >
> > > We hope these address your remaining concerns, and that you would consider your score further.
> > >
> > > Reference:
> > > - Shwartz-Ziv, R., and Armon, A. (2022). Tabular data: Deep learning is not all you need. Information Fusion, 81, 84-90.

---

### Official Review · Reviewer_qrNr · 2025-03-04

**Significance:** 4
**Argument Clarity:** 3
**Rating:** 4
**Confidence:** 4

**Questions:**

I do not have any specific question. This is a good paper in my stack, except for some few typos which can be eliminated after a proof reading exercise.

**Discussion Potential:**

3

**Paper Summary:**

The paper focuses on evaluating the behaviour of 6 recent, scalar, generative fidelity and diversity pairs of metrics, namely I-Prec and I-Rec (Kynkaanniemi et al, 2019), Density and Coverage (Naeem et al, 2020), $\alpha$-precision and $\beta$-recall (Alaa et al, 2022), C-Prec and C-Rec (Khayatkhoei & Abdalmageed, 2023) and P-Prec and P-Rec (Park & Kim, 2023). To evaluate the metrics, the authors proposed a list of 6 desiderata (sec 2) defining what we want from synthetic data metrics, and came up with 6 groups of sanity checks (in total of 30 sanity checks) revolving simple distributions, geometries in low and high dimensions and tabular data (sec 3). These sanity checks are mapped to the desiderata, showing that they map to all but the last desiderata about computational efficiency, which the present metrics satisfy. Results show that all fidelity and diversity metrics fail multiple sanity checks, leading to the paper's proposition that all current fidelity and diversity metrics are flawed. The paper then presents takeaways for researchers and practitioners, and give arguments for alternative views.

## update after rebuttal

My review was already very positive before the rebuttal. I've kept my score.

**Position:**

Yes

**Position In Title:**

Yes

**Related Work:**

4

**Strengths And Weaknesses:**

This is a well-written paper with all the necessary materials to support the stated position are presented. The fidelity and diversity metrics are evaluated with a comprehensive set of sanity checks which I found reasonable. The success criteria for these sanity checks (app C) are practically convincing, meaning that the findings presented in the 2 main tables (Table 3 & 4) are reliable. As research in generative modelling is an active topic these days, I think this paper will inspire discussion and encourage the community to research for new and better metrics than existing metrics. For example, the paper has pointed out that all of the evaluated metrics use Euclidean distances which can fail to distinguish things in Euclidean geometry. The authors arguments over most common Alternative Views are reasonable to me. Recommendations in app A are very useful to the reader.

**Support:**

4

---

> ### Author Rebuttal · Authors · 2025-03-31
>
> Thank you for the review. We will proofread the paper and fix any typos during the revision. Please let us know if you have any other suggested improvements.

---

### Official Review · Reviewer_TDnJ · 2025-03-11

**Significance:** 3
**Argument Clarity:** 3
**Rating:** 3
**Confidence:** 4

**Questions:**

- I agree the main message although I am not an expert/active researcher on the evaluation aspect of generative models. To clarify [W1], the below are questions that I have concerned:
    - [1] Positioning two direct related work (Borji 2019 and Xu et al 2018) is required to clarify why the authors started this position paper from Section 2 in the current format.
    - [2] Do desiderate (D1-D6) have some priority order? (Is D1 prior to D2?), which should be clarified and explained.
    - [3] If possible, please add more explanations on D1(c); I can partially follow your paragraph (l.113-120), but not fully understand this one.
    - [4] Are sanity checks are straightforwardly developed or defined? Did the authors develop them or arrange them from existing studies? I got lost when I entered in Sec 4, making me feel that the structure here is a bit ad-hoc.
- Furthermore, as noted in Section 5.3, "It is not our intention to claim that a metric must pass all of our checks in order to be useful. When I first read the paper, I wonder if there is such a perfect metric. I understand 'we need to consider metrics carefully', but in our pessimistic perspective we may not have such a perfect metric and therefore we need to design some principles (which may be a point made by the paper). Do the authors have any ideas in this direction?

**Discussion Potential:**

3

**Paper Summary:**

This is a position paper on how the two evaluation axes (fidelity and diversity) of generative models have been used and what properties they have had in previous research. Based on previous research (Table 1), the authors organized six desiderata and constructed (a list of) sanity checks (Table 2), from the revision of previous evaluation criteria discussed in recent research and related to the desiderata, and through implementation and experimentation, we presented two takeaway messages (one for practitioners, the other for researchers).

## update after rebuttal

The authors have constructively clarified my concerns, and their comments are valuable to say the importance of the topic. Here, I keep my positive score after the rebuttal.

**Position:**

Yes

**Position In Title:**

Yes

**Related Work:**

3

**Strengths And Weaknesses:**

**Strenghts**
- [S1] Well-structured and well-organized paper, particularly positioning the authors’ consideration into existing metrics.
- [S2] Related to [S1], a good list of links to resources and related work (e.g., B) with explicitly stated takeaway messages.

**Weakness**
- [W1] An ad-hoc (or, scattered) introduction of the discussion contents: desiderata and metrics.

**Support:**

3

---

> ### Author Rebuttal · Authors · 2025-03-31
>
> Thank you for the review.
>
> > [W1] An ad-hoc (or, scattered) introduction of the discussion contents: desiderata and metrics.
>
> > [1] Positioning two direct related work (Borji 2019 and Xu et al 2018) is required to clarify why the authors started this position paper from Section 2 in the current format.
>
> Can you clarify this, we are not sure how additional discussion on Borji (2019) and Xu (2018) would clarify why we started with our desiderata list in Section 2?
>
> The reason we started with the desiderata list is that to us it seems natural to start a paper investigating how well evaluation metrics work by defining what properties a good metric has. Also, Section 3 has references to the desiderata in the embedding discussion, so if we changed the order of these sections, we would need to move the embedding discussion elsewhere.
>
> > [2] Do desiderate (D1-D6) have some priority order? (Is D1 prior to D2?), which should be clarified and explained.
>
> They do not have a priority order, except as explained in the text for D6 (computation). We will clarify this in the revision.
>
> > [3] If possible, please add more explanations on D1(c); I can partially follow your paragraph (l.113-120), but not fully understand this one.
>
> We will rewrite this explanation to only state why the target metric should be impractical for a proxy metric to make sense, and not include the statement that previous works have done unmeaningful evaluations. We think this will clarify the point, and also fix an issue with the specific example we had that was pointed out by reviewer QPmc.
>
> > [4] Are sanity checks are straightforwardly developed or defined? Did the authors develop them or arrange them from existing studies? I got lost when I entered in Sec 4, making me feel that the structure here is a bit ad-hoc.
>
> Most of the sanity checks are from previous literature, and a handful were developed by us. These are specified in Table 2, and detailed descriptions of the checks are in Appendix C, which are referenced in the opening paragraph of Section 4. We will clarify this paragraph to make these references more prominent.
>
> > ​​Furthermore, as noted in Section 5.3, "It is not our intention to claim that a metric must pass all of our checks in order to be useful. When I first read the paper, I wonder if there is such a perfect metric. I understand 'we need to consider metrics carefully', but in our pessimistic perspective we may not have such a perfect metric and therefore we need to design some principles (which may be a point made by the paper). Do the authors have any ideas in this direction?
>
> We think there is definitely a chance that a perfect metric does not exist, since perfect things rarely exist in general. Your point about the need to design principles is what we are saying with the “metrics need to be considered carefully” takeaway. If a perfect metric does not exist, or is not currently known, one must sacrifice something when choosing a metric, so one should know what they are choosing to sacrifice when choosing a metric.

---

### Official Review · Reviewer_Rb5W · 2025-03-16

**Significance:** 2
**Argument Clarity:** 2
**Rating:** 2
**Confidence:** 3

**Questions:**

1. Why do the authors choose to focus only on tabular data?
2. How can more general claims be made from the more specific analyses presented?

**Discussion Potential:**

2

**Paper Summary:**

The paper takes the position that current metrics ("fidelity" and "diversity") for assessing data synthesized using generative models are flawed and the development of new metrics is the need of the hour instead of building new models. To assess this, a list of desiderata and automatic sanity checks are designed and evaluations are considered on synthetic test beds.

**Position:**

Yes

**Position In Title:**

Yes

**Related Work:**

2

**Strengths And Weaknesses:**

**Strengths:**
- The paper presents a systematic and potentially useful automated mechanism for metric evaluation for tabular data.

**Weaknesses:**
- The motivation and takeaways are presented to be more general than the setting of tabular metrics that are, in fact, the focus of the position.
- The evaluations are only presented on synthetic datasets (as is also acknowledged in Section 5.3, though it is unclear to me whether it correctly belongs in that section). To judge potential impact, I believe the authors must present case studies on real-world datasets and clearly demonstrate current lacunae and the efficacy of their diagnostics.

**Typos:**
- L253: success **or** failure
- L306: **upper bounds**

**Support:**

2

---

> ### Author Rebuttal · Authors · 2025-03-31
>
> Thank you for the review.
>
> > The motivation and takeaways are presented to be more general than the setting of tabular metrics that are, in fact, the focus of the position.
>
> > Why do the authors choose to focus only on tabular data?
>
> We do not focus only on tabular data. The sanity checks we include from literature focusing on image data represent a majority of the checks, so our takeaways apply outside tabular data. The word “tabular” in the title of Section 2 was left over by mistake from an earlier draft, and will be removed in the revision.
>
> > The evaluations are only presented on synthetic datasets (as is also acknowledged in Section 5.3, though it is unclear to me whether it correctly belongs in that section). To judge potential impact, I believe the authors must present case studies on real-world datasets and clearly demonstrate current lacunae and the efficacy of their diagnostics.
>
> > How can more general claims be made from the more specific analyses presented?
>
> We only use simple data, since with real data, we would not be able to focus on a single potential flaw as easily as with simple data, and we would not know the values the metrics should have precisely enough to evaluate D4 (bounds), as we mentioned in Section 5.3. Failures in these checks are relevant to real data, since the potential flaws they investigate can happen in real data, and we find it unlikely that the additional complexities that real data bring would somehow fix flaws that a given metric has with much simpler data.

---

### Decision · Program_Chairs · 2025-04-30

**Decision:**

Accept (poster)

**Comment:**

I recommend accepting this submission. The reviewers generally found the paper well-structured and comprehensive, with praise for the systematic evaluation framework and practical recommendations. There was significant debate about whether this qualifies as a position paper versus original research; I agree with the eventual consensus that position papers can be strengthened by experimental results. Reviewers also raised concerns about the lack of real-world dataset experiments and some checks being too specific to tabular data. The paper's core message and evaluation was widely recognized as valuable, and the authors addressed most reviewer concerns in their rebuttal.